

# Automated snow cover detection on mountain glaciers using space-borne imagery

Rainey Aberle[1], Ellyn Enderlin[1], Shad O'Neel[2], Caitlyn Florentine[3], Louis Sass[4], Adam Dickson[1], Hans-Peter Marshall[1], Alejandro Flores[1]

[1]Department of Geosciences, Boise State University, Boise, ID 83725, USA
[2]Cold Regions Research and Engineering Laboratory, U.S. Army Corps of Engineers, Hanover, NH 03755, USA
[3] U.S. Geological Survey, Northern Rocky Mountains Science Center, West Glacier, MT, USA
[4] U.S. Geological Survey, Alaska Science Center, Anchorage, AK 99508, USA

*Correspondence to*: Rainey Aberle (raineyaberle@u.boisestate.edu)

**Abstract.** Tracking the extent of seasonal snow on glaciers over time is critical for assessing glacier vulnerability and the response of glacierized watersheds to climate change. Existing snow cover products do not reliably distinguish seasonal snow from glacier ice and firn, preventing their use for glacier snow cover detection. Despite previous efforts to classify glacier surface facies on local scales, a unified approach for monitoring glacier snow cover on larger spatial scales remains elusive. We present an automated snow detection workflow for mountain glaciers using supervised

machine learning-based image classifiers and Landsat 8/9, Sentinel-2, and PlanetScope satellite imagery. We develop the image classifiers by testing numerous machine learning algorithms with training and validation data from the U.S. Geological Survey Benchmark Glaciers. The workflow produces daily to biweekly time series of several glacier mass balance and snowmelt indicators (snow-covered area, accumulation area ratio, and seasonal snowline) from 2013 to present. Workflow performance is assessed by comparing automatically classified images and snowlines to manual interpretations at each glacier site. The image classifiers exhibit overall accuracies of 92–98%, Kappa scores of 84–

96%, and F-scores of 93–98% for all image products. The median difference between automatically and manually delineated median snowline altitudes, along with the interquartile range, averages 27 +/- 79 m across all image products. The Sentinel-2 classifier (Support Vector Machine) produces the most accurate glacier mass balance and snowmelt indicators and distinguishes snow from ice and firn the most reliably. Yet, the Landsat- and PlanetScope-

derived estimates greatly enhance the temporal coverage and frequency of observations. Additionally, the transient accumulation area ratio produces the least noisy time series, providing the most reliable indicator for characterizing seasonal snow trends. The temporally detailed accumulation area ratio time series reveal that the timing of minimum snow cover conditions varies by up to a month between Arctic (63° N) and mid-latitude (48° N) sites, underscoring the potential for bias when estimating glacier minimum snow cover conditions from a single late-summer image.

Widespread application of our automated snow detection workflow has the potential to improve regional assessments of glacier mass balance, water resources, and the impacts of climate change on snow cover across broad spatial scales.

## 1 Introduction

Glaciers in Alaska and the western United States and Canada lost 267 ± 6 Gt of mass between 2000 and 2019, more than 25% of the global mass lost from glaciers outside the ice sheets (Hugonnet et al., 2021). The recent accelerated

glacier mass loss in western North America is well correlated with changes in regional precipitation and summer air



temperature (Hugonnet et al., 2021; O'Neel et al., 2019), indicating that decreased snow accumulation was likely an important driver. Furthermore, North American snow water resources are in steep decline (Musselman et al., 2021). Projections suggest ongoing decreases in annual snow water equivalent across the western contiguous United States (Siirila-Woodburn et al., 2021) and Alaska (Littell et al., 2018) throughout the 21st century.

The decline in snow water resources directly impacts glacier surface mass balance, the balance between snow accumulation and ablation. Time series of snow-covered area (SCA) on glaciers can be used as a first-order indicator of glacier mass balance (Cuffey & Paterson, 2010). Several SCA-derived metrics are commonly used to assess glacier health, including the accumulation area ratio (AAR), or the fraction of the total glacier area that is covered by snow at the end of the summer melt season, and the altitude of the seasonal snowline at the end of the annual melt season,
often used to estimate the equilibrium line altitude (ELA). However, observations of snow cover distribution for glaciers across western North America remain sparse (Huss et al., 2014; D. McGrath et al., 2017), underscoring the need for an automated remote sensing approach to address this gap.

Snow extent is often mapped in satellite images using the Normalized Difference Snow Index (NDSI), which differentiates snow, ice, and firn from other earth surface materials and clouds by leveraging their distinct contrast in
visible and short-wave infrared (SWIR) reflectance (Hall & Riggs, 2007). Firn is snow that has persisted through at least one melt season and is therefore typically darker than and visibly distinct from bright snow. Figure 1a depicts spectral signatures for several earth surface types. The NDSI is calculated as:

$$NDSI = \frac{\rho_G - \rho_{SWIR}}{\rho_G + \rho_{SWIR}} \tag{1}$$

where $\rho_G$ and $\rho_{SWIR}$ are the reflectance values of the green and SWIR bands, respectively (Dozier, 1989). The NDSI
has been used to quantify SCA and fractional snow cover on non-glacier surfaces using various satellite platforms such as MODIS (Salomonson & Appel, 2004), Landsat (Riggs et al., 1994), and Sentinel-2 imagery (Gascoin et al., 2019) with an NDSI threshold of about 0.4 (Dozier, 1989; Hall & Riggs, 2007; Sankey et al., 2015). However, each of these satellites has inherent tradeoffs between spatial and temporal resolution. High temporal resolution is particularly important in mountainous areas where cloud probability can exceed 50% (Gascoin et al., 2015; Parajka &
Blöschl, 2008), leading to more unusable images. While MODIS provides global repeat imagery every one to two days, its coarse spatial resolution (250–1000 m) limits its ability to resolve changes in SCA at scales smaller than 1 km². In contrast, Sentinel-2 provides imagery at a resolution of 10–20 m, but on an approximately five-day repeat basis in the absence of cloud cover.

Mapping changes in SCA on mountain glaciers over time remains a challenge in part due to the relatively small size
of glaciers, the similar spectral characteristics of snow, ice, and firn, as well as the fact that the SCA can change rapidly near the end of the summer melt season, when SCA observations provide critical constraints on glacier surface mass balance. A substantial portion of glaciers in the western United States and Canada, 11% by area and 82% by number according to the Randolph Glacier Inventory (RGI Consortium, 2017), has an area of less than 1 km². This size constraint hinders the use of satellite images with spatial resolutions of 1 km or more for mapping snow cover on these



glaciers. Several recent studies have worked to overcome gaps in spatial and temporal coverage of SCA estimates associated with image repeat intervals, cloud cover, and spatial resolution. Techniques such as data fusion and spatial downscaling (Berman et al., 2018; Rittger et al., 2021; Vincent, 2021; Walters et al., 2014), leveraging multiple satellite image products (e.g., Gascoin et al., 2019), and the use of the 3–5 m-resolution ~daily PlanetScope imagery (Cannistra et al., 2021; John et al., 2022) have helped to overcome these gaps. Yet, applying the NDSI thresholding

method to glacier surfaces is likely to yield inaccurate results due to the overlapping NDSI ranges for snow versus ice and firn. Figure 1d shows the SCA classified using the standard NDSI threshold of 0.4 for a Landsat image over Wolverine Glacier, which classifies most to all bare ice and firn on the surface as snow. Although seasonal snow exhibits a slightly higher NDSI value than ice and firn in the image depicted in Figure 1c, the NDSI values for each surface type vary seasonally and from year to year. This variability limits the use of an NDSI threshold alone for

accurately distinguishing snow from ice and firn. As such, there is a need for a snow classification method specifically calibrated to glacier surfaces in order to accurately map their SCA variability in space and time.

**Figure 1: a) Spectral signatures for various Earth surface types and corresponding band ranges for different satellite image product resolutions: PlanetScope 4-band 3–5 m resolution, Sentinel-2 10 m and 20 m resolution, and Landsat 8/9 30 m**

**resolution. Thermal bands, which are beyond the range of spectral signature observations, have been excluded. The spectral signatures are adapted from Painter et al. (2009) for snow, from Salvatori et al. (2022) for ice and firn, and from the USGS Spectral Library Version 7 (Kokaly et al., 2017) for vegetation, soil, and seawater. b) Landsat 8 image captured on 17 August 2017 at Wolverine Glacier, Alaska, with snow covering the northern section of the image and bare ice and rock/vegetation exposed in the southern section. c) Resulting snow classification from the same Landsat 8 image illustrating**

**an example of erroneous classification of bare ice as snow using the standard NDSI threshold of 0.4, where blue represents pixels classified as snow (NDSI >= 0.4) and white represents areas classified as no snow (NDSI < 0.4).**

Our goals in this work are two-fold: (1) Adapt components of previous snow remote sensing techniques, including

leveraging multiple space-borne image products, the use of newly available PlanetScope imagery, and machine





learning, to develop an automated snow detection workflow calibrated to glacier surfaces, and (2) identify the optimal
imagery dataset(s) and snow cover metrics for assessing spatiotemporal trends in glacier snow cover.

Below, we describe the approach to address these goals, including the study sites used to construct the training and
validation datasets (2.1), image pre-processing steps (2.2), the classification models development (2.3), construction
of the glacier snow cover time series (2.4), and performance assessment of the image classifiers (2.5). We then outline
results for the performance assessment (3.1) and evaluate patterns and seasonality of the snow cover time series at the
U.S. Geological Survey (USGS) Benchmark Glaciers (3.2). In the Discussion, we outline limitations of the workflow
(4.1), assess which image product and snow cover metric derived from the workflow produces the most robust glacier
snow cover time series (4.2–4.3), and finally, outline the broader impacts and potential applications of the workflow
(4.4).

## 2 Methods

We developed an automated snow detection workflow using supervised machine learning (ML) models and Landsat
8/9, PlanetScope, and Sentinel-2 satellite image products. The validation data were constructed for five glaciers in
North America that sample mid- and high-latitudes and maritime and continental climate regimes (Fig. 2a). To assess
the performance of each ML image classifier, we compared automated snow-covered area (SCA) maps to a separate
validation dataset comprised of manually generated snow cover observations at all glacier sites. The image classifiers
were then used to construct SCA maps at each site for 2013–2023. From the SCA maps, we extracted time series of
the accumulation area ratio (AAR) and the median snowline altitude. The transient AAR offers a normalized
representation of glacier SCA with respect to its total area over time. At the end of the melt season, the AAR provides
insights into the fraction of the glacier area with positive surface mass balance. The study sites selected for workflow
development and testing as well as detailed descriptions of the workflow steps are described in the subsections below.

### 2.1 Study sites

The five mountain glaciers in this study are part of the U.S. Geological Survey (USGS) Benchmark Glacier Project,
which began in 1957 (Meier, 1958) as part of a long-term initiative to document and understand connections between
glaciers and climate (O'Neel et al., 2019). The project includes seasonal field and remote sensing data collection of
glacier mass balance at five glacier sites located across the western contiguous U.S. and Alaska. These glaciers have
diverse characteristics, such as aspect, latitude, continentality, and elevation (Fig. 2), all of which influence local
climate regime and glacier mass balance. Gulkana Glacier is located in the Alaska Range with an elevation range of
1235–2445 m, Wolverine Glacier is located in the Kenai Mountains of Alaska (472–1673 m), Lemon Creek Glacier
is located at the southernmost tip of the Juneau Icefield in Alaska (663–1500 m), Sperry Glacier is located in Glacier
National Park in Montana (2274–2791 m), and South Cascade Glacier is located in the northern Cascade Range in
Washington State (1635–2204 m). Multitemporal boundaries and digital elevation models (DEMs), constructed using
Maxar stereo satellite imagery and the Ames Stereo Pipeline (Shean et al., 2016), are available at each site from the
USGS (McNeil et al., 2022).



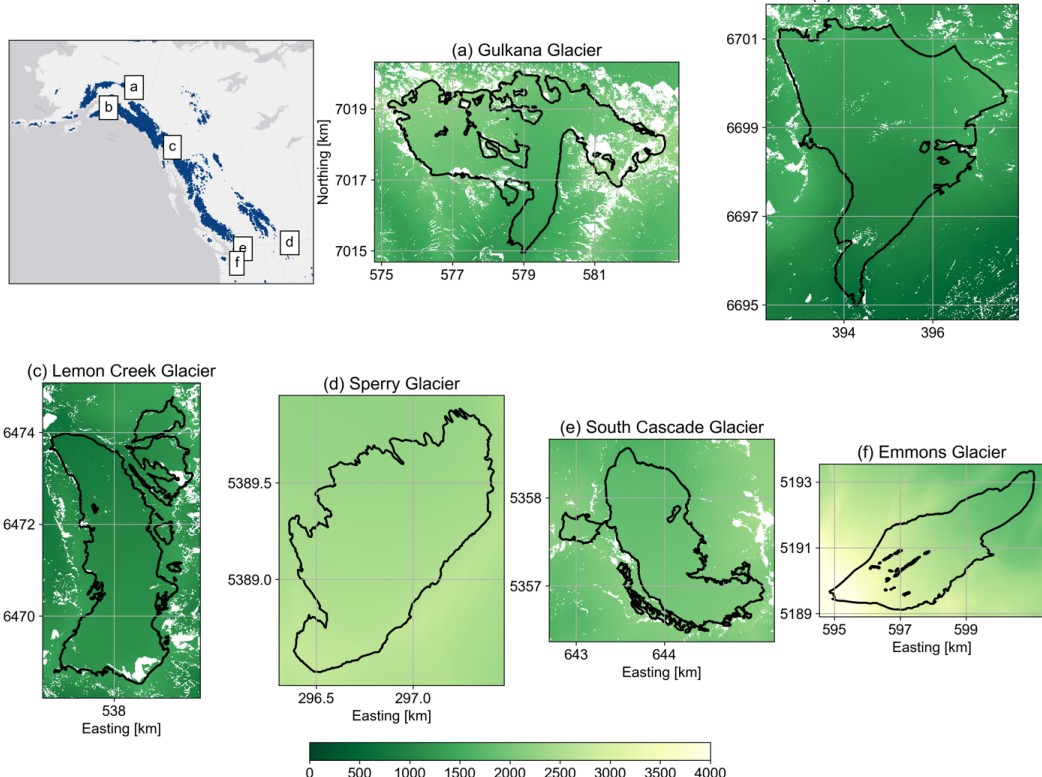

**Figure 2: Maps of the study sites in order of decreasing latitude. (Upper left) Locator map of study sites with all identified**
**glaciers in the region shaded in blue (Randolph Glacier Inventory regions 1 and 2; RGI Consortium, 2017). Background**
**image is from Esri (2015). (a)–(f) Digital elevation models (DEMs) and glacier boundaries outlined in black for the USGS**
**Benchmark Glaciers and Emmons Glacier in Washington State. Note that Emmons Glacier was only included in the**
**performance assessment and not the training dataset for the image classifiers. For the Benchmark glaciers, DEMs and**
**glacier boundaries are from the USGS data release version 8 (McNeil et al., 2022) for the most recent date. For Emmons**
**Glacier, the DEM is from the NASADEM (NASA JPL, 2020), and glacier boundaries are from the Randolph Glacier**
**inventory version 6 (RGI Consortium, 2017). Coordinates are with respect to the local UTM zone at each site.**

**2.2 Image pre-processing**

To optimize temporal coverage of glacier SCA time series, we developed the snow detection workflow using Landsat
8/9, PlanetScope, and Sentinel-2 imagery. The characteristics of each satellite image product are listed in Table 1. We
preferentially selected surface reflectance (SR) products rather than top-of-atmosphere reflectance (TOA) products,
because the atmospheric corrections on SR products generally allow for better change detection on the Earth's surface
(Masek et al., 2006). However, because Sentinel-2 SR imagery is only available since 2018, we also included all
Sentinel-2 TOA products available since 2015 to increase temporal coverage. Additionally, even though the
PlanetScope images used in this analysis were harmonized with Sentinel-2 as the target sensor (Planet Labs, Inc.,
2022), we still found a wide distribution of dynamic ranges between images. For example, the maximum SR values
for two images at the same site captured under similar conditions are 0.8 and 1.5. To better unify the imagery, we



developed an additional pre-processing step for PlanetScope imagery shown in Figure 3 and described in the supplement (S1). Briefly, the median SR values in the highest elevation portions of each glacier were assumed to consist primarily of snow, such that the SR dynamic range was adjusted to vary from zero (for the darkest pixels) to
0.94 for the blue band, 0.95 for the green band, 0.9 for the red band, and 0.78 for the near infrared band, based on SR of fresh snow from Painter et al. (2009). Although this adjustment relies on the potentially biased assumption that the upper elevations contain fresh snow, we found that this step improved the overall accuracy of PlanetScope snow detection.

**Table 1: Data products used in the automated snow detection workflow.**

| Dataset (Sponsoring organization) | Spatial resolution (m) | Temporal resolution | Temporal Coverage | Spectral range (μm) | Orbit altitude (km) |
|---|---|---|---|---|---|
| Landsat 8-9 Surface Reflectance OLI/TIRS (NASA, USGS)[1,2] | 30/100 | Bi-weekly | 2013–present | 0.43–12.51 | 705 |
| PlanetScope 4-band Surface Reflectance (Planet Labs, Inc.)[3] | 3–5 | ~Daily | 2016–present | 0.47–0.89 | 450–525 |
| Sentinel-2 Top-of-Atmosphere Reflectance (ESA)[4] | 10–60 | Weekly | 2015–present | 0.49–13.75 | 786 |
| Sentinel-2 Surface Reflectance (ESA)[4] | 10–60 | Weekly | 2018–present | 0.49–13.75 | 786 |
| Site-specific data: glacier boundaries and digital elevation models[5] | ~2 m, ~2 m | Annual | 1950–2021, varying by site | N/A | N/A |

[1]U.S. Geological Survey (2013); [2]U.S. Geological Survey (2022); [3]Planet Labs, Inc. (2022); [4]European Space Agency (2015); [5]McNeil et al. (2022)

### 2.3 Classification models development

In recent decades, ML models have been increasingly used for land cover classification (Thanh Noi & Kappas, 2018).
ML models exhibit exceptional proficiency in handling multidimensional data (such as many image bands) and complex class characteristics (Maxwell et al., 2018), potentially making them an ideal tool for distinguishing snow from other surface types in glacierized environments. Supervised ML models, which require user-constructed training data, tend to outperform unsupervised models in land cover classification applications (Bahadur K. C., 2009; Boori et al., 2018). To apply this approach for snow detection, we developed the ML models by first constructing a large
validation dataset at the USGS Benchmark Glaciers and Emmons Glacier, Washington. Emmons Glacier was only included in the performance assessment and not the training dataset for the image classifiers to provide a more challenging test case for the classification workflow, as described below. We then trained nine supervised ML models, including the foremost models for land cover classification (Thanh Noi & Kappas, 2018; Wang et al., 2021), using a subset of the validation dataset. The best model for each image product was determined using K-folds cross-validation
(Hastie et al., 2009). To evaluate the models' transferability to other sites and to assess potential model overfitting, we conducted the performance assessment at two sites not included in the model training data.

To construct the validation dataset, we manually classified more than 8,000 points for each image product (~32,000 points total) at the USGS Benchmark Glaciers as snow, shadowed snow, ice/firn, rock, or water in several images at each site. We also explored the inclusion of a dedicated firn class that was distinct from the ice/firn class. To develop



this dedicated firn class, we used manually classified points at Wolverine Glacier, where firn is clearly visible on the surface late in the melt season most years. However, we found that including the dedicated firn class increased misclassifications and that the performance of the supervised ML snow detection workflow was superior using the ice/firn class. Classified points for Lemon Creek and Emmons Glaciers were set aside for the classification performance assessment (Section 2.4), while classified points from all other sites were combined to construct the

training dataset. We consider Lemon Creek Glacier a relatively ideal site for classification given its simple geometry, limited topographic shading, and typically continuous ice and snow on the surface without obstruction from things like debris or surface water (Fig. 2c). The Benchmark Glaciers sample diverse climatic regimes and extensive latitudinal range, but these sites do not represent the full suite of complexity in glacier surface types, due to limited debris cover and shaded areas. To ensure that the classification workflow could handle sites that we consider more

complex than the Benchmark Glaciers for classification, we also set aside manually classified points at Emmons Glacier in Washington State. Emmons has more frequent topographic shading, extensive debris cover, and intermixed surface types particularly late in the melt season (Fig. 2f). Images at each site were selected to span different months of the melt season in an effort to capture a wider distribution of reflectance values for each class (i.e., surface type).

       Points in each image were chosen using stratified random sampling (Cochran, 1977) such that the number of points

chosen for each class was roughly proportionate to the areal coverage (Fig. 3). In an effort to minimize bias associated with user interpretation, we only selected points where we could confidently interpret the surface type. As a consequence, we avoided points close to the snow-firn or snow-ice boundary where the class distinction was unclear. Only a small fraction of the points classified as ice were clearly firn, given the difficulty in interpretation in most cases. At each classified point, all visible, infrared, and thermal band values were sampled from the respective satellite

image. Next, we calculated the NDSI for each point (Eqn. 1). Because PlanetScope does not include a SWIR band, the NDSI for each PlanetScope pixel was calculated using the green and near-infrared bands instead of the green and SWIR bands. The normalized difference of the green and NIR bands theoretically captures the distinct signatures of snow, ice, and firn compared to other materials, similar to that of the green and SWIR bands in other image products (Fig. 1a).



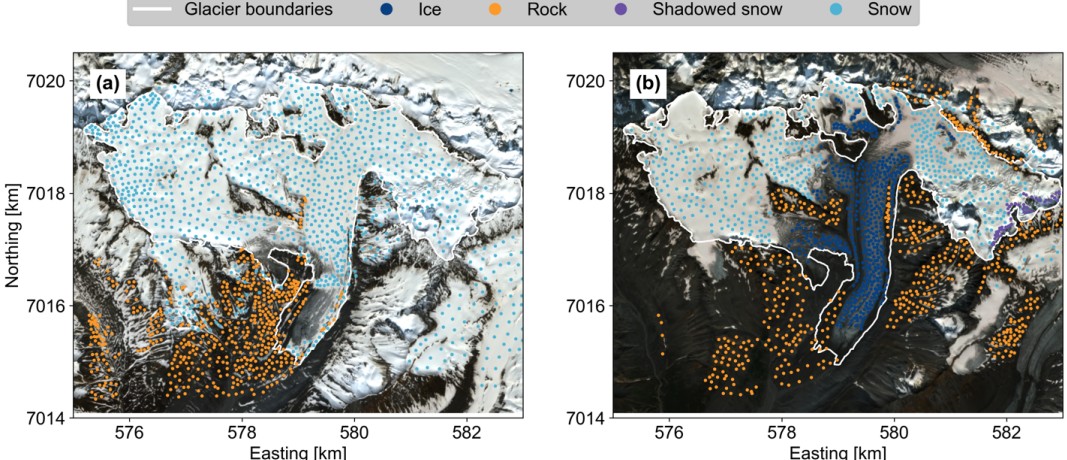

**Figure 3. Example manually classified points used to construct the training dataset at Gulkana Glacier, Alaska. Points are shown for two Sentinel-2 surface reflectance images captured a) 2021-06-15 and b) 2021-08-06. Coordinates are with respect to UTM zone 6N.**

For each image product, we tested nine supervised ML models: AdaBoost, Decision Tree, Naïve Bayes, Nearest

Neighbors, Neural Network, Random Forest, Support Vector Machine, Quadratic Discriminant Analysis, and Logistic Regression. While the Support Vector Machine, Random Forest, Nearest Neighbors, and Neural Network models are generally reported to be foremost models for land cover classification (Thanh Noi & Kappas, 2018; Wang et al., 2021), we tested several others because our classes are unique from typical land cover classification applications. ML models were accessed through the Python-based ScikitLearn toolbox (Pedregosa et al., 2011). Due to the unique band

characteristics of each image product, a separate model was required for each product (Fig. 4). To train and assess each ML model using K-folds cross-validation, the training dataset was split into ten equally sized subsets, or "folds," the model was trained using nine folds, then tested on the remaining fold, iterating this process until all folds were set aside and used to calculate the overall accuracy. The model with the highest mean overall accuracy was determined the optimal model for each image product and retrained using the full training dataset. To investigate the robustness

of our model selections, we also calculated learning curves for each ML model, which provide insight into the dependence of a model's performance on the training dataset size (Viering & Loog, 2023).

**2.4 Method application: snow cover detection**

To capture the evolution of glacier snow cover throughout the ablation season, we applied the pretrained ML models to satellite images collected from 1 May to 1 November (Fig. 4, step 2), which encompasses all 21st century minimum

mass balance dates recorded for the USGS Benchmark Glaciers (McNeil et al., 2022). The time series spans 2013 to 2023 based on image availability (Table 1). Images were clipped to the closest glacier boundary in time (McNeil et al., 2022) and masked for clouds, heavy haze, and cloud shadows using each image product's respective cloud mask. A data table showing the timestamps of each glacier boundary and DEM for each snow detection year is provided in the supplementary material (S3, Table S2). While the cloud masks are subject to occasional errors, particularly for





PlanetScope, we found that large clouded areas were typically identified well by the cloud masks and using them to mask images improved the time series of classified images overall. Additionally, users have the option to turn off cloud masking using the `mask_clouds` parameter in the workflow to evaluate the performance of cloud masking products for a specific site. To maximize spatial coverage, all images captured within the same hour by the same satellite, typically consisting of images with distinct but overlapping footprints, were used to construct a median

composite image (Fig. 4, top panel). Through the process of mosaicking, the images were spatially aligned, and any overlapping pixels were filled with the median value of each band. All images with less than 70% coverage of the glacier area were then removed from the collections. After testing a number of thresholds, we found that the 70% threshold sufficiently filtered very cloudy and hazy images, while preserving the highest number of clear, usable images for our study sites. Nonetheless, we suggest testing different thresholds in the workflow using the

`cloud_cover_max` and `aoi_coverage` parameters when applying the workflow to other sites. The classification models were then applied to their respective image collections, resulting in classified image collections at each site. The SCA for each image was calculated as the total number of snow-covered pixels (both "snow" and "shadowed snow" classes) within the glacier boundary, discarding masked pixels, and multiplied by the appropriate pixel resolution. The AAR was calculated using the ratio of the SCA to the total glacier area.

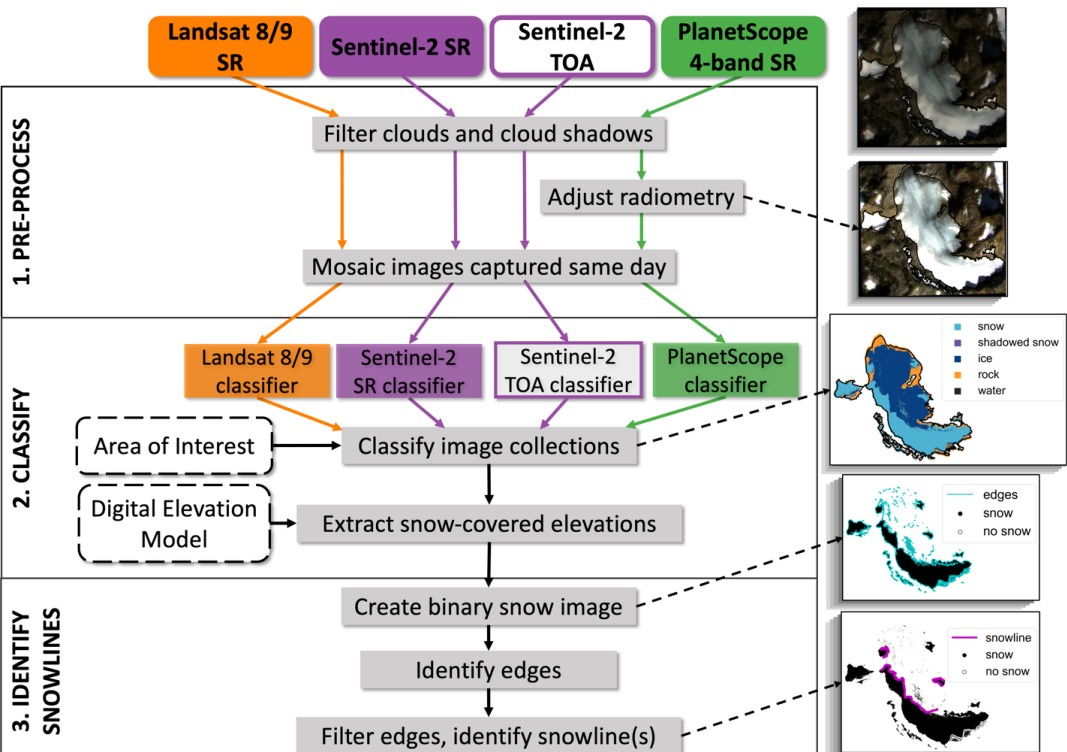


**Figure 4: Schematic of the image processing workflow. Images on the right show example results for a PlanetScope image captured at South Cascade Glacier, Washington State, on 24 September 2021, where dashed black lines indicate the corresponding processing step.**



Next, seasonal snowlines were automatically identified by adjusting and analyzing the classified images (Fig. 4, step
3). To mitigate the impact of small topographic features (e.g., exposed bedrock, crevasses, or small patches of snow)
on the detection of the seasonal snowline, classified images were adjusted using the distribution of snow-covered
pixels and the boundaries between snow and no-snow ("edges"). Assuming that elevation is a primary driver of snow
cover distribution and glacier mass balance (e.g., Anderson et al., 2014; Cuffey & Paterson, 2010; McGrath et al.,
2018), we first filled holes in the SCA maps using an elevation histogram-based approach. For each glacier, we
generated a reference elevation histogram for the entire glacier area using the closest high-resolution (~2 m, Table 1)
DEM in time. Next, we generated histograms of snow-covered elevations for each SCA map with 10 m-increment
bins spanning the glacier elevation range. A normalized histogram representing the percentage of each elevation bin
covered in snow was constructed by dividing the snow-covered elevations histogram by the glacier elevations
histogram. After testing a number of thresholds, all elevation bins with at least 75% snow coverage were set to 100%,
and the classified image pixels were adjusted accordingly. Next, we created binary snow masks ("snow" or "no snow")
and identified the edges between snow and no-snow using an edge detection function from the scikit-image package
(Walt et al., 2014) built on the marching squares algorithm (Lorensen & Cline, 1987). The no-data mask for each
image was then buffered by 30 m (the coarsest image spatial resolution) to remove edges identified at data boundaries.
Edge points within the no-data mask were removed, edges with gaps spanning more than 100 m were split into separate
edge segments, and edge segments with total lengths less than 100 m were removed, resulting in the final snowline(s).
Elevations were then extracted from the DEM for each snowline vertex coordinate, which were used to track the
distribution of snow-covered altitudes and the median snowline altitude over time. The final snowlines consist of
coordinates at the spatial resolution of the input image.

In summary, the workflow produces classified image collections at the spatial resolution of each input image and data
tables containing statistics for each classified image and snowline. Statistics include the SCA, AAR, snowline
coordinates, surface elevations at each snowline coordinate, and the median snowline altitude.

## 2.5 Performance assessment

### 2.5.1 Snow cover classification

To evaluate the performance of the ML models, we applied the classification algorithm to two glaciers that were
excluded from the model training process: Lemon Creek and Emmons Glaciers. The classified maps for Lemon Creek
and Emmons Glaciers were compared to the validation set of approximately 5,000 manually classified points. To
focus on the accurate mapping of SCA on the glaciers, the validation points were classified as either "snow" (snow
and shadowed snow classes) or "no snow" (including ice, firn, water, and rock/debris). Validation points were selected
using stratified random sampling, similar to the training points selection described in Sect. 2.3.

Each image classifier's performance was assessed using the overall accuracy, Cohen's Kappa score, recall, precision,
and F-score (or F1 score) metrics. To calculate these metrics, we sampled the SCA maps generated by the ML models



at each manually classified point, assuming the manually classified points to be ground truth. The overall accuracy is the portion of correctly classified pixels (Campbell & Wynne, 2011):

$$Overall\ accuracy = \frac{True\ positives + True\ negatives}{Number\ of\ samples} \tag{2}$$

Cohen's Kappa score ($K$) accounts for potential random agreement between the classified image and the validation points:

$$K = \frac{Observed - Expected}{1 - Expected} \tag{3}$$

where *Observed* is the overall accuracy and *Expected* is the correct classification due to chance (Cohen, 1960). The Kappa score ranges from -1 to +1, with positive values indicating that the trained model performs better than a random
model. For example, a random classification model with two classes (e.g., "snow" and "non-snow") would have an *Expected* overall accuracy of 0.5 (50%). If the accuracy of the trained model, or *Observed* accuracy, is 0.85 (85%), this would result in a Kappa score of 0.7.

Recall generally indicates the classifier's ability to identify all positive ("snow") samples:

$$Recall = \frac{True\ positives - True\ negatives}{True\ positives + True\ negatives} \tag{4}$$

Precision represents the classifier's ability to not label a negative sample as positive (i.e., not to label "non-snow" as "snow"):

$$P = \frac{True\ positives}{True\ positives + False\ positives} \tag{5}$$

For example, consider a binary classification problem where 30 pixels are snow and 70 pixels are non-snow. If the model classifies 20 pixels as snow, out of which 15 are actually snow (true positives) and 5 are non-snow (false
positives), and the remaining 10 pixels snow are incorrectly classified as non-snow (false negatives), this would result in a *Recall* of 0.6 (60%) and a *Precision* of 0.75 (75%). These metrics indicate that the model identified 60% of all snow pixels in the dataset, and when the model classified snow pixels, it was correct 75% of the time.

Finally, F-score reflects both false positives and false negatives of the classifier using the harmonic mean of precision and recall:

$$FScore = 2 * \frac{Precision * Recall}{Precision + Recall} \tag{6}$$

The *FScore* ranges from 0 to 1, with higher values indicating both high precision and high recall, giving equal weight to precision and recall. Given a *Recall* of 0.6 and a *Precision* of 0.75 as above, the *FScore* would be 0.67.



### 2.5.2 Snowline detection

To assess the performance of the automated snowline detection method, we used cloud-free PlanetScope imagery to manually delineate snowlines at each Benchmark Glacier for approximately five dates per year spanning the summer melt season for 2016–2022. The manually delineated snowlines were then interpolated to ensure equal spacing with a ground resolution of 30 m, the coarsest satellite image resolution. For each pair of manually and automatically delineated snowlines, we calculated the distance between each coordinate of the manually delineated snowline and the nearest corresponding point on the automatically detected snowline (i.e., ground distance) and the difference in
median snowline altitude between the manually and automatically delineated snowlines.

Field-based annual ELA estimates that are independent of imagery and the supervised ML approach are available for the USGS Benchmark Glaciers. However, direct comparison with our snowline altitudes is hindered by methodological differences. Specifically, the USGS calculates ELAs as the elevation where surface mass balance is equal to zero according to a piecewise linear regression fit to in situ point measurements of mass balance. In situ
measurements are collected on field campaigns that target favorable weather windows near the annual mass minimum, typically in August or September of each year (O'Neel et al., 2019). However, the USGS method lacks control for ELA estimates that extend beyond the glacier's elevation range. For example, this can occur in a strong negative balance year, when the fitted gradient approach will extrapolate the equilibrium balance altitude above the glacier, resulting in ELA estimates that surpass the actual elevation limits of the glacier even for years where a small
accumulation zone on the glacier surface persists. Consequently, such estimates fall outside the bounds of our snowline delineation method, presenting a methodological artifact that may or may not be a robust representation of the real-world conditions. Therefore, we focused our assessment on comparing the automatically and manually detected snowlines. Nonetheless, our image-based snow cover estimates may be used to help constrain ground-based ELA estimates by the USGS and other communities.

### 3 Results

### 3.1 Performance assessment

Here, we outline the performance assessment of the optimal image classification models. Assessment of all other tested machine learning (ML) models (n=9) are presented in the supplement (Table S1). Results for the snow cover classification and snowline detection performance with respect to manual, image-based observations using the optimal
image classification algorithms are shown in Table 2.

Each ML model exceeded 83% values across all performance metrics. The optimal models are the Nearest Neighbors model for the Landsat and PlanetScope SR image products and the Support Vector Machine model for the Sentinel-2 SR and TOA image products. All classification models have an estimated overall accuracy of at least 92%, a Cohen's Kappa score greater than 83%, and an F-score of at least 93%. The Sentinel-2 SR Support Vector Machine classifier
performs best according to the performance metrics, with an overall accuracy of 98%, a Kappa score of 96%, and an



F-score of 98%. The Landsat classifier is the least accurate of all optimal image product classifiers, yet it still yields an overall accuracy of 92%.

The learning curves analysis revealed that varying the training dataset between 500 and 6,500 sample points did not meaningfully change which model was the most accurate and led to minimal fluctuations (±5%) in cross-validated accuracy scores for the optimal models. This consistent performance instils confidence that the optimal models are not prone to overfitting and are robust for all training dataset sizes greater than ~1,500 points. The learning curves for all models are shown in the supplementary material (S2, Fig. S2).

Automatically detected snowlines differ from manually delineated snowlines by a median (+/- interquartile range) of 116 +/- 239 m in ground distance and -28 +/- 66 m in median elevation. Thus, the automatically detected snowlines tend to be slightly lower in elevation than the manually delineated snowlines. Sentinel-2 SR and TOA automatically derived snowlines are the closest to the manually delineated snowlines, with median differences in elevation of -6 +/- 40 m and -17 +/- 54 m, respectively. In terms of image pixels, Sentinel-2 automatically derived snowlines are within 5 +/- 17 and 5 +/- 19 pixels of manually delineated snowlines, respectively. While Landsat yielded the highest disagreement in terms of ground distance, it yielded a median distance of 8 +/- 13 pixels from manually delineated snowlines, much better than PlanetScope, which had median pixel distances of 52 +/- 79 pixels. Potential explanations for the varying agreement between manually and automatically detected snowlines for each image product are discussed in Sect. 4.1.

**Table 2: Performance of the snow detection workflow for each satellite image product. SR indicates surface reflectance and TOA indicates top-of-atmosphere reflectance. Snowline statistics indicate the median +/- the interquartile range of differences for all dates and sites tested, where negative differences indicate that the automated snowline estimates are lower in elevation than the manually delineated snowlines. Median ground distances for automatically detected snowlines are also reported with respect to distance in pixels (px.) for each image product.**

| Image product | Snow classification | | | | | | Snowline detection | |
| | Optimal classification model | Overall accuracy | Cohen's Kappa score | Recall | Precision | F-score | Median ground distance [m] (pixel distance) | Difference in median elevation [m] |
|---|---|---|---|---|---|---|---|---|
| Landsat 8-9 SR | Nearest Neighbors | 92 % | 84 % | 91 % | 94 % | 93 % | 211 +/- 375 (8 +/- 13 px.) | -60 +/- 116 |
| PlanetScope 4-band SR | Nearest Neighbors | 96 % | 91 % | 97 % | 94 % | 96 % | 154 +/- 236 (52 +/- 79 px.) | -30 +/- 53 |
| Sentinel-2 SR | Support Vector Machine | 98 % | 96 % | 97 % | 99 % | 98 % | 50 +/- 158 (5 +/- 17 px.) | -6 +/- 40 |
| Sentinel-2 TOA | Support Vector Machine | 93 % | 87 % | 89 % | 99 % | 94 % | 49 +/- 185 (5 +/- 19 px.) | -17 +/- 54 |
| All image products average | N/A | 95 % | 89 % | 93 % | 97 % | 95 % | 116 +/- 239 (18 +/- 32 px.) | -28 +/- 66 |

### 3.2 Snow cover timeseries

Figure 5 shows the weekly median trend and interquartile range in snow-covered area (SCA), normalized snow-covered area (i.e., the transient accumulation area ratio; AAR), and median snowline elevation from the Sentinel-2- and Landsat-derived observations for the full 2013–2023 time series. We focus on Sentinel-2- and Landsat-derived snow cover time series because the PlanetScope-derived observations were much noisier, leading to less interpretable



median trends in each snow cover metric. Complete SCA time series including PlanetScope observations are shown
in the supplementary material (Fig. S3).

In general, the transient AAR time series suggest that the largest, most northerly sites, Gulkana and Wolverine
Glaciers, have higher annual AARs compared to other sites. Transient AARs for both Wolverine and Gulkana Glaciers
vary from nearly the entire glacier extent (~1) in May and October to a seasonal minimum of about 0.6 +/- 0.3 in
August. While the glacier areas are much smaller at Lemon Creek, South Cascade, and Sperry Glaciers, resulting in
much lower ranges in seasonal SCA, the ranges in transient AARs are much larger (0.1–0.3) than Wolverine and
Gulkana Glaciers. The minimum transient AAR consistently occurs in August or early September at Lemon Creek
Glacier and in late September or October at Sperry and South Cascade Glaciers.

On average, the onset of seasonal snow accumulation (i.e., rapid increase in the SCA) varies between sites. The decline
and recovery of snow cover typically happen earliest in the year at Gulkana Glacier compared to the other sites (Fig.
5a–c). Here, the transient AAR typically declines in June, reflecting the exposure of bare ice and decline in snow
cover, and reaches a minimum of about 0.6 in August. The transient AAR then increases between August and October,
signaling the onset of snowfall. At Wolverine Glacier, the transient AAR both declines and recovers one to two weeks
later than at Gulkana on average. The transient AAR time series for Lemon Creek Glacier (Fig. 5g–i) indicates that
bare ice exposure likely also begins in May at this site but that minimum snow cover (transient AAR of 0.5 +/- 0.4) is
reached in either August or September. In contrast, for both South Cascade and Sperry Glaciers (Fig. 5j–o), which are
both located at lower latitudes and higher average elevations than the other sites, bare ice is not exposed until June or
July, with a faster decline and lower end state AAR of ~0.3–0.5 in September. While these trends in SCA decline and
recovery broadly correlate with site latitude, they are likely also related to other factors such as climate and elevation,
as discussed further in Sect. 4.






**Figure 5: Weekly median trends in snow cover metrics. The first column shows snow-covered area (SCA; blue), the second column shows transient accumulation area ratio (AAR; yellow), and the third column shows median snowline altitude at each site for the full 2013–2022 time series excluding PlanetScope observations to reduce noise (pink). Solid lines indicate the weekly median value and shaded regions indicate the weekly interquartile range. Dashed gray lines in all panels indicate maximum glacier area in the first column and the minimum and maximum glacier elevations in the third column.**




## 4 Discussion

Leveraging multiple satellite image products and supervised machine-learning algorithms, we produce detailed time
series of seasonal changes in glacier snow cover. Below, we discuss limitations of the workflow associated primarily
with topographic shading, cloud cover, and firn exposed on the surface. Moreover, we outline how the Sentinel-2 SR-
derived observations and the transient AAR effectively overcome these limitations more consistently than other image
products and metrics derived from the workflow. Finally, we analyze the SCA time series at the USGS Benchmark
Glaciers, highlighting variations in the timing of the snow ablation season and snow distribution patterns between
sites and their implications for glacier mass balance studies.

### 4.1 Limitations

Complex topography, such as steep ridges along the glacier margin, can lead to less accurate SCA results due to the
misclassification of shadowed snow. Sperry Glacier, for example, is particularly challenging due to its frequent
topographic shading that covers a large portion of the glacier. When shade is cast by the glacier's southeastern ridge,
the shaded region within the SCA is more likely to be misclassified as ice or rock. The shadowed snow class mitigates
this instance of misclassification in some images (Fig. 6e–f), but not others (Fig. 6a–b). Additionally, non-continuous
snowlines, characterized by large patches of bare ice within the SCA, are common at Sperry Glacier later in the melt
season, which can lead to varied detection of the snowline. In these cases, the snowline may be detected within the
SCA rather than at the lowest altitude boundary separating snow and ice. Alternatively, the snowline may be divided
into small segments less than 100 m in length that are filtered out before the final snowline selection. Similarly,
Gulkana Glacier has multiple tributaries that flow into the main trunk. When the snowline rises above the convergence
of these tributaries, the snowline may be detected in one branch but not the others, depending on the image quality
and length of each snowline segment (Fig. 6a–b, g–h). In the case of multiple glacier tributaries and consistently
patchy snow cover distribution, using the SCA and AAR time series to assess snow cover trends may be especially
beneficial.





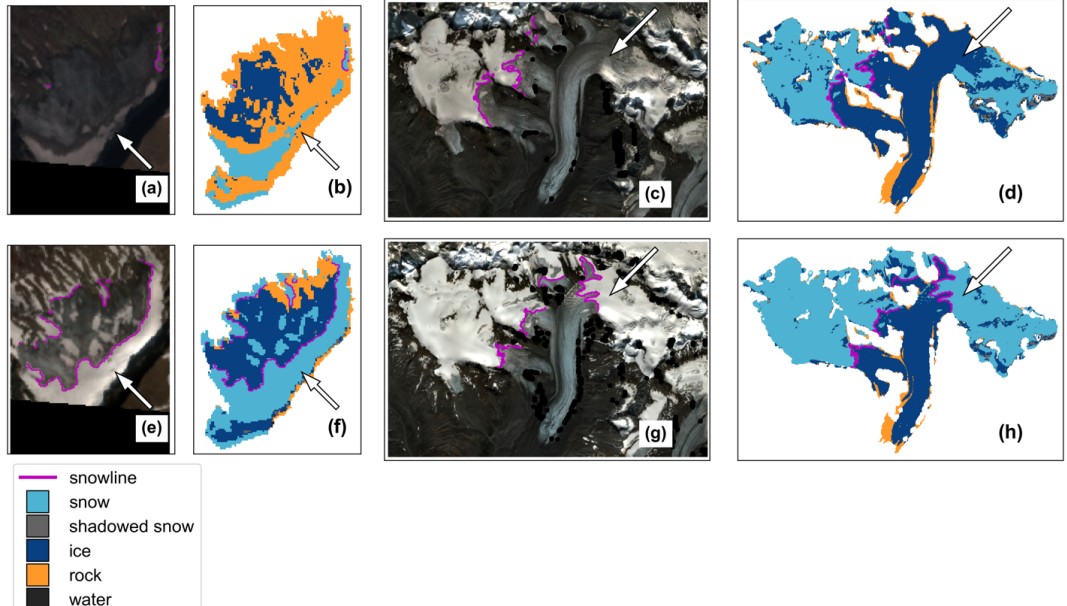

**Figure 6: RGB and classified images demonstrating potential limitations of the snow detection workflow under different challenging scenarios. a)–b) and e)–f) At Sperry Glacier, topographic shading and patchy snow cover can lead to shadowed snow being misclassified as rock, with no snowline detected in some instances. c)–d) and g)–h) At Gulkana Glacier, multiple glacier tributaries can complicate snowline detection, with varying degrees of success in different tributaries.**


Certain geographic regions, particularly coastal maritime sites, will have more frequent cloud cover throughout the melt season. More frequent cloud cover will result in sparser SCA time series due to either an abundance of cloudy images that are automatically filtered from the image collection or masking clouds detected in large portions of a given

image. Because clouds, haze, and cloud shadows are masked in each image, the SCA and/or snowline may be fully or partially masked. The location of the site with respect to satellite ground tracks can also impact SCA accuracy. Particularly for sites that sit between satellite path boundaries with minimal overlap, the site area may exceed the coverage of images captured within the same hour, leading to consistently incomplete SCA estimates.

The distinction between the ice-firn and firn-snow boundaries poses challenges for glacier mass balance studies,

including the automated snow detection workflow developed in this study. We explored the inclusion of a firn class in the classifiers using manually classified points at Wolverine Glacier, where firn is known to exist on the surface late in the melt season for most years. However, the addition of the firn class resulted in a decrease in classifier performance across all metrics due to an increase in misclassifications. We observed that the dedicated firn class was particularly sensitive to image illumination. For example, a relatively dark image would lead to most seasonal snow

being incorrectly classified as firn. Given the similarity in spectral signatures among firn, snow, and ice—especially during the late melt season when seasonal snow has a lower albedo and dust/debris are more prevalent—there is substantial overlap in the spectral signatures of the training data when firn is added as a distinct class. Consequently, our classified images and snowline estimates in absence of the firn class occasionally detect the firn-snow boundary or the ice-firn boundary, depending on factors such as ground conditions, image illumination, and the presence of





clouds or haze. Rather than introduce a firn class, we suggest preferentially selecting, or heavily weighting, Sentinel-2 SR-derived observations at sites where firn is known to be exposed on the surface because Sentinel-2 SR-derived classified images tend to distinguish firn from snow better than other image products. Figure 7 shows an example pair of images where the firn is problematically classified as snow in Sentinel-2 TOA imagery (Fig. 7a–b) and the firn is correctly classified as ice (not snow) in Sentinel-2 SR imagery (Fig. 7c–d) captured on the same date.

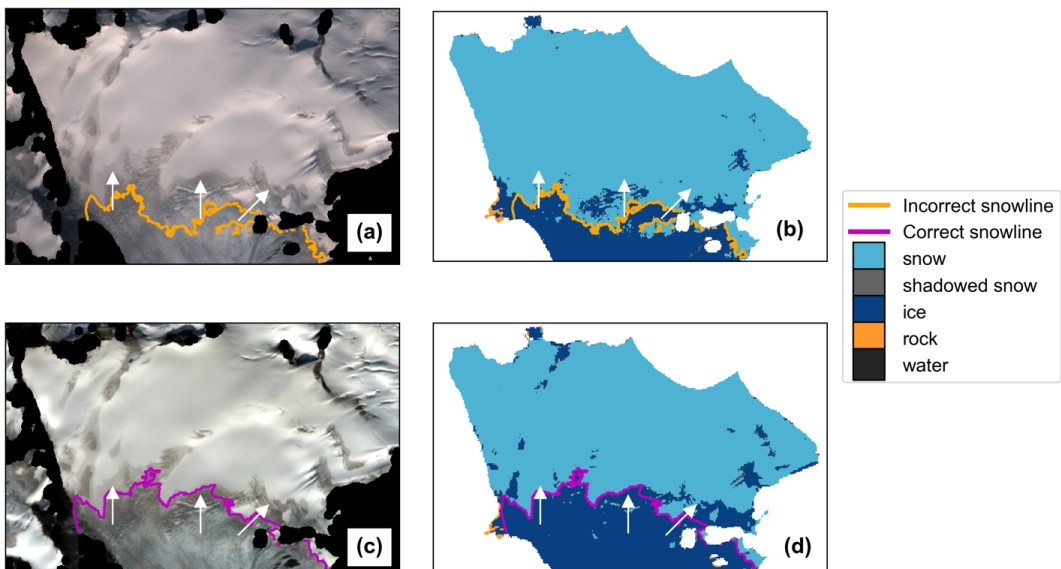


**Figure 7: Example snow detection results at Wolverine Glacier demonstrating how the Sentinel-2 SR-derived snow maps more reliably capture the snow-firn boundary than that of other image products. RGB and classified images captured 22 August 2020 from a)–b) Sentinel-2 TOA and c)–d) Sentinel-2 SR. White arrows point to firn exposed on the surface.**

The potential impact of misclassification of firn as snow on SCA analyses will vary with the abundance of firn exposed
during particularly high melt or low snow years, which depends on glacier geometry and firn extent. For instance, surface slope can substantially impact how much firn may be exposed during a relatively high melt year. Consider a 20 m rise in the snowline altitude for two glaciers of equal width, one with a constant slope of 10° and another with a constant slope of 20°. The shallower, 10°-sloped glacier will have a greater area of exposed bare ice and/or firn on the surface (see S4, Fig. S4 for a more detailed example and diagram). In this case, particularly for the shallower-sloped
glacier, we suggest Sentinel-2 SR-derived observations be weighted more heavily. Thus, the choice of image products to use when applying the workflow may depend on the specific glacier surface characteristics.

## 4.2 Optimizing temporal density and spatial coverage for surface mass balance assessment

High temporal resolution and coverage of SCA estimates, particularly near the end of the melt season, are critical for accurate constraints on glacier surface mass balance. While Sentinel-2 stands out as the preferred choice for generating
SCA time series due to its relatively smooth time series, optimal tradeoffs between spatial and temporal resolution, and consistent snow and ice/firn discrimination, other image products contribute valuable observations.



PlanetScope has dense temporal coverage and produces classified images with overall accuracies comparable to Sentinel-2 and Landsat. However, SCA time series produced with PlanetScope imagery are 'noisy,' meaning there is considerable scatter between observation dates relative to the Sentinel-2 and Landsat time series (Fig. S3), due to the lower image quality (i.e., differences in reflectance between images for the same earth material), cloud masking product limitations, and a narrower spectral range of PlanetScope images. The lower orbit altitude (Table 1) and lower quality cameras compared to the governmental satellite constellations contribute to occasional saturation and less reliable cloud masks, introducing uncertainties in SCA estimates of an unknown amount. Despite efforts to normalize reflectance values between images (Sect. S1), the limited spectral range of PlanetScope imagery, particularly at wavelengths beyond the near infrared, and its frequently saturated image bands limit its ability to distinguish snow from other surface types (see Fig. 1a).

Landsat, despite its sparse bi-weekly revisit time, provides valuable observations before 2016. However, the true minimum snow cover conditions may not be captured with Landsat observations alone. The longer revisit time relative to Sentinel-2 and PlanetScope, combined with frequent late summer cloud cover particularly in maritime regions, sometimes results in entire melt seasons without usable images. Challenges in time series interpretation due to the lower temporal resolution of Landsat images is pronounced for smaller glaciers like Sperry Glacier, where the likelihood of completely masked images due to cloud cover is higher.

Sentinel-2 excels in overall performance, yet observations from PlanetScope and Landsat substantially extend and add detail to the SCA time series. The unique strengths and weaknesses of each image product highlight the need for a thoughtful integration strategy to provide comprehensive insights into glacier snow cover dynamics. By combining these satellite image products, we create a robust dataset essential for comprehensive glacier mass balance studies.

### 4.3 Snow cover metrics comparison

While cloud cover can introduce noise into the SCA time series, the transient AAR is less impacted by cloud masking and produces the least noisy time series overall. Using the ratio of SCA to the total glacier area, the calculation of the AAR effectively counteracts the impact of heavily masked images. The automated snowline delineations can be biased by small, isolated patches or "edges" of disconnected snow that can skew the median snowline altitude. In contrast, the AAR captures snow cover at the scale of the glacier area and is therefore less sensitive to small patches of snow. In Figure 5, the transient AAR has the lowest variability in weekly values compared to the other metrics in the early melt season, when rapid changes in snow cover are unexpected.

Additionally, the AAR does not assume that a single elevation contour on the glacier represents the zero-mass balance line, unlike the median snowline altitude or ELA. The use of a mass balance indicator with minimal assumptions is particularly important for sites where shading or other topographic effects exert a stronger influence on snow cover distribution than elevation alone. At South Cascade Glacier, for example, the steep headwalls along the southwestern boundary serve as avalanche source regions in the winter and also provide topographic shading throughout the ablation season (O'Neel et al., 2019; Fig. 8a). As a result, the automatically detected snowline in the late melt season crosses





a wide range of elevations (1788–2085 m; Fig. 8b). Therefore, using the median snowline altitude as an indicator of snow cover decline and recovery rather than the transient AAR or distribution of snow-covered elevations could miss important topographic or other climate controls on snow distribution over time.

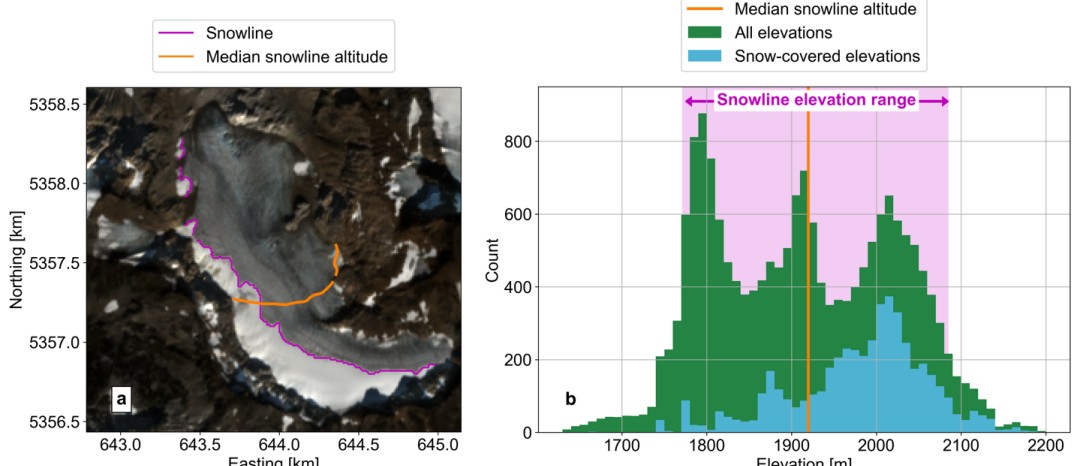

**Figure 8: a) Sentinel-2 surface reflectance image at South Cascade Glacier captured 2 October 2023, with the automatically detected snowline and the median snowline altitude contour. b) Histograms of all elevations in the glacier area (green) and snow-covered elevations (blue). The orange line indicates the automatically detected median snowline altitude and the shaded maroon region shows the full elevation range (1788–2085 m) of all snowline coordinates.**

Notably, the AAR accuracy depends on a time-evolving glacier boundary (Florentine et al., 2023). Outdated boundaries can lead to misleading results in cases where the glacier area has changed in response to climate or internal dynamics. For instance, a few years of lower-than-average snowfall or higher-than-average melt can lead to glacier thinning and terminus retreat over several years (Cuffey & Paterson, 2010). If glacier boundaries are not updated over time as in this study, the AAR will be underestimated. Thus, glacier boundaries should be updated as needed when applying the workflow.

## 4.4 Strengths and future work

The automated snow detection workflow offers substantial time savings compared to manual snowline delineation. Implementing the full workflow for all satellite image products from 2013–present at one site, including PlanetScope image downloads and pre-processing, typically requires an hour or less from the user and anywhere from about 2–50 hours of computation time, depending on the computing resources, size of the site, and number of images found. In comparison, manual delineation of glacier snowlines can take approximately 1–5 minutes per image, with additional time needed for PlanetScope image downloads. Considering that our automated method identified an average of ~750 usable images per site since 2013, manually delineating all snowlines for a single site would require more than ~20 hours of work by the user. Adopting our automated method could save hundreds of hours for the user, particularly when applying it to multiple sites, making it an efficient approach for tracking changes in glacier snow cover on broad spatial scales.



Despite potential limitations related to shading, cloud masking, and firn misclassification affecting snow detection at individual sites, the extensive time series of image-based snow cover observations generated by this workflow holds promise for various applications. These observations can improve our understanding of current glacier AARs and ELAs, supporting glacier-climate sensitivity tests. For example, the dense time series of snow cover observations can

help to constrain the timing of minimum snow cover conditions. Notably, for the Benchmark Glaciers, snow-off conditions tend to occur later (June–July) at South Cascade and Sperry Glaciers. Sperry and South Cascade Glaciers are the lowest in latitude, with Sperry Glacier located at the highest elevations and South Cascade Glacier located in a mid-elevation, maritime climate. Snow-on conditions tend to occur earlier (September–October) at the Alaskan glaciers, which span maritime and continental climates and low to mid-elevations. These findings demonstrate the

valuable insights gained into the spatiotemporal variability of snow cover minimum conditions across latitudinal, climatic, and elevational ranges spanned by the Benchmark Glaciers through the application of the automated snow detection workflow. Additionally, these snow cover observations can serve as inputs or observational constraints for climate modeling, offering valuable validation data for snowmelt and atmospheric modeling applications, thereby advancing our understanding of diverse Earth system interactions.

**5 Conclusions**

In this study, we present an automated snow detection workflow calibrated to mountain glaciers, offering several advantages over existing methods for snow classification. Our approach leverages multiple space-borne imagery datasets, resulting in hundreds of snow cover observations spanning over a decade. Temporal resolutions range from approximately biweekly to daily throughout the summer melt season, depending on local cloud cover conditions and

generally increasing over time with the launch of additional satellites. In future work, we will apply the automated snow detection workflow more broadly to glaciers throughout North America. Additionally, the workflow may be tested in other climatic settings, such as tropical or polar glacierized environments. This would enable us to evaluate the transferability of the classification models, particularly in light of potentially distinct spectral responses of snow, ice, surface meltwater, and debris in these regions.

Using a training dataset constructed at the USGS Benchmark Glaciers and supervised machine learning models, the image classification models exhibit high performance, achieving overall accuracies of at least 92%. Kappa scores, which account for potential correct classification due to chance, range from 84–96% for all classification models. The workflow performance and temporal coverage are impacted by a number of factors, primarily the presence and frequency of widespread shading and cloud cover at individual sites.

Among the image classification models, the Sentinel-2 surface reflectance (SR) classification model produces the most accurate and smoothest snow-covered area time series (overall accuracy > 95%), with the best agreement with manually delineated snowlines (median altitudes within -6 +/- 40 m). Sentinel-2 SR classified images also distinguish snow from ice and firn the most consistently at our study sites. Therefore, we suggest weighting Sentinel-2 SR-derived observations more heavily than that from other image products particularly at sites where extensive firn is known to



be exposed on the surface. Nonetheless, Landsat- and PlanetScope-derived observations greatly increase both the temporal coverage and frequency of observations, which are critical near the end of the melt season when snow cover changes rapidly.

Furthermore, our results reveal variations in the timing of bare ice exposure and snowfall onset across the USGS Benchmark Glaciers. The observed spatial variations in minimum SCA for this subset of glaciers emphasizes that
estimating the equilibrium line altitude (ELA) based on a fixed date, such as late September, can lead to biased results depending on the glacier site. Additionally, non-elevation-dependent snowlines at South Cascade Glacier, in particular, challenge the assumption of a single ELA as an accurate approximation of the accumulation and ablation zone boundary on mountain glaciers.

The automated snow detection workflow has the potential to benefit numerous scientific disciplines and applications.
By improving our understanding of glacier snow dynamics, such as snow distribution, accumulation, and ablation patterns, we not only enhance glacier monitoring but also provide a validation dataset for atmospheric, hydrological, and glacier modeling. The insights gained from our approach therefore have the potential to improve the accuracy of climate model predictions, guide water resource management, and refine our understanding of evolving snowmelt seasons.

**Data and code availability**

Landsat and Sentinel-2 images were accessed through the Google Earth Engine (GEE) data repository (https://developers.google.com/earth-engine/datasets). PlanetScope images were downloaded through the Planet Labs, Inc., Python API (https://developers.planet.com/docs/apis/). Glacier boundaries and DEMs are from the USGS Benchmark Glacier Project's most recent data release (McNeil et al., 2022). Classified images and snow cover metrics
for the U.S. Benchmark Glaciers for 2013–2023 are available through the National Snow and Ice Data Center (pending publication). All code used for method development and application are available via Zenodo (Aberle et al., 2024) and as a public GitHub repository (https://github.com/RaineyAbe/glacier-snow-cover-mapping). U.S. Geological Survey research is supported by the Ecosystem Mission Area Climate R&D Program.

**Competing interests**

The contact author has declared that none of the authors has any competing interests.

**Acknowledgments**

Any use of trade, firm, or product names is for descriptive purposes only and does not imply endorsement by the U.S. Government. This work was funded by BAA-CRREL award W913E520C0017, NASA EPSCoR award 80NSSC20M0222, the NASA Idaho Space Grant Consortium summer internship program, and the SMART (Science,
Mathematics, And Research for Transformation) scholarship program. This study used data made available through



the NASA Commercial Smallsat Data Acquisition (CSDA) Program. The authors acknowledge constructive reviews from Taryn Black (NASA Goddard) and Tanner May (University of Michigan) that improved the quality of this manuscript.

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
