# Peer review of "Automated snow cover detection on mountain glaciers using space-borne imagery and machine learning"

_EGUsphere, 2024_

## Referee Comment (RC1)

**Summary:**

The manuscript *Automated snow cover detection on mountain glaciers using space-borne imagery* presents an analysis of machine learning (ML) models and  workflow developed by the authors for automatically detecting snowlines on mountain glaciers in Western N. America, using optical imagery from Landsat, PlanetScope and Sentinel-2. After testing nine different ML models, support vector and nearest neighbor were the optimal algorithms for classifying Sentinel-2 and Landsat/PlanetScope imagery, respectfully. The authors report a high accuracy assessment using four different accuracy metrics. Overall, their paper provides a novel insight into optimal models for different image types and the usefulness of ML for glacier applications.  I would recommend minor revisions before publication.

**General Comments:**

A strength of the paper is highlighting the accuracy of the various ML outputs and identifying which ML model works for each imagery dataset. Furthermore, the robustness of testing and validating the models allows for high confidence in the performance of the models. The authors take care to select training data from a variety of sites with varying climate, surface type and geometries. The paper is well written and organized, making it easy to follow the decisions and outcomes of the papers. The novelty of the paper comes out in the testing of several ML algorithms for snow line detection.

Snowline detection is an illustrious metric within glaciology for first order estimates of surface mass balance and for understanding melt extents/intensity. Published research in this area of remote sensing has been somewhat extensive but that is not reflected in the introduction or discussion of the paper. The authors should rewrite the introduction and discussion to reference relevant literature on machine learning and snowline detection. Currently, there is a good background on the use of NDSI in mapping snowlines, but it is missing the context of previous work on snow line detection through ML and optical imagery. Furthermore, the discussion would benefit from framing of previous studies and how their results compare with other work.

The methods section gives a good overview of each step and is nicely summarized in Figure 4. I would like to see additional sections that expands on the machine learning models, giving a brief background on the nine different types and why they are selected. For instance, identifying which models are ensemble tree types, kernel type, neural type. Perhaps, highlighting which models have been previously applied in glacier applications.

The authors conclude the manuscript with stating that the performance of the ML model depends on the image product and specific characteristics of the glacier site, which is valuable contribution to the field on snowline detection in mountain environments. The concluding remarks are supported by the main results and justify the methodology used.

**Specific (minor) comments:**

Title – Consider including machine learning and optical imagery in the title, such as, 'Automated snow cover detection on mountain glaciers using optical imagery AND machine learning'.

Figure 1 – Include caption for Figure (d).

Figure 2 – Include grid or scale bar/northing arrow and land boundary or better contrast in colors between land and ocean in the location image. Increase font size/bold labels.

Line 150 – Figure 3 → should be Figure 4 and order of figures needs to be reconsidered within the text.

Line 165 – Can you include the process of training and validation dataset development within figure 4? Or a separate figure to have a visual summary of how the datasets were developed.

Line 165 – This contradicts Line 270 where it mentions that Lemon Creek Glacier was used in the performance assessment, should that also be mentioned here?

Line 167 – Consider listing the nine ML models here or state that they will be listed at the end of this section.

Line 169 – Include number of folds here.

Line 172 – 188 – This paragraph could be better clarified to first talk about the validation dataset and then what's used for training. Some of the information is contained within the preceding paragraph, so it is difficult to know if it is a repeat or new information.

Line 189 – Did this result in an even number of points for each class? If not, how do you reconcile the bias within classes that are under sampled?

Line 204 – Was this at all sites for each image product?

Line 215 – Mention that this is expanded on in supplementary material.

Line 247 – Clarify that you are discussing situations for missing data.

Line 343 – Refer to either Table 2 or Figure 7 to where to look for the results of changes in snow line detection.

Line 360 – SCA has been defined earlier.

Line 366 – Refer to figure 5 (a-f) here, as is done in the next paragraph.

Line 401 – Consider renaming this section to 'Challenges in Classification' or something similar, since limitations are not necessarily being discussed here.

Line 505 – Readers should be referred to Table S2 for specifics on when glacier boundaries were updated.

Figure 6 – Include in caption what the white arrows are highlighting.

Line 510 – This section does not mention future work. Consider changing title or including future work.

Line 570 – Mention in the methods section that images were accessed with GEE.

**Supplemental material:**

Line 601 – Check figure numbering, Figure S4 → S3

---------------------------------------------------------END OF REVIEW---------------------------------------------------------

---

## Author Comment (AC1)

We thank the reviewers for their thoughtful comments and suggestions for the manuscript. Below, we outline our plans for revision based on the comments from each reviewer. Reviewer comments are shown in blue and our responses are shown in black for clarity.

**Review 1**

Snowline detection is an illustrious metric within glaciology for first order estimates of surface mass balance and for understanding melt extents/intensity. Published research in this area of remote sensing has been somewhat extensive but that is not reflected in the introduction or discussion of the paper. The authors should rewrite the introduction and discussion to reference relevant literature on machine learning and snowline detection.

We agree that more background on glacier snow classification and machine learning in particular is needed in the Introduction. To better couch this work in the literature related to classifying glacier snow cover with a focus on machine learning, we will replace paragraph 3 of the Introduction (L48–L63) with a discussion of the following approaches:

- Thresholding techniques
    - The Normalized Difference Snow Index (NDSI) has been used to quantify SCA and fractional snow cover on non-glacier surfaces using various satellite platforms such as MODIS (Salomonson & Appel, 2004), Landsat (Riggs et al., 1994), and Sentinel-2 imagery (Gascoin et al., 2019) with an NDSI threshold of about 0.4 (Dozier, 1989; Hall & Riggs, 2007; Sankey et al., 2015).
    - Otsu thresholding (Otsu, 1979), an automated threshold selection approach for gray-level images, has been used to detect glacier snowlines using Landsat 8 imagery in Switzerland (Prieur et al., 2022) and the Austrian Alps (Rastner et al., 2019).
- Machine learning techniques
    - Neural Networks have been applied to PlanetScope imagery in the alpine western U.S. (Cannistra et al., 2021; John et al., 2022)
    - Random Forest has been applied to Sentinel-2 imagery in Alaska (Zeller et al., In review)
    - Support Vector Machine has been applied to C-band SAR imagery (Callegari et al., 2016; Huang et al., 2013; Li et al., 2012)

In paragraph 4, we will replace the discussion of NDSI limitations (L74–L79) with a focus on the lack of previous work that directly compares machine learning models applied to various image products for classifying snow and ice.

We will also modify panels c and d of Figure 1 to better support these changes. Instead of the NDSI-based classification panels, we will present box plots of band reflectance and NDSI values throughout a given melt season for a sample snow- and ice-covered pixel. We hope this will better illustrate one of the main challenges of automated snow and ice classification: the overlapping and temporally evolving reflectance values of snow and ice.

The methods section gives a good overview of each step and is nicely summarized in Figure 4. I would like to see additional sections that expands on the machine learning models, giving a brief background on the nine different types and why they are selected. For instance, identifying which models are ensemble tree types, kernel type, neural type. Perhaps, highlighting which models have been previously applied in glacier applications.

Thank you for the suggestion. To provide more context for each of the machine learning models tested, we will edit L204:

"For each image product, we tested nine supervised ML models, including linear (Logistic Regression, Nearest Neighbors), quadratic (Quadratic Discriminant Analysis), non-parametric (Decision Tree), kernel-based (Support Vector Machine), ensemble (AdaBoost, Random Forest), Naïve Bayes, and Neural Network models."

We will also highlight which have been previously used in glacier snow classification as you suggested in the Introduction, mainly the Support Vector Machine (Li et al., 2012), Random Forest (Zeller et al., In review), and Neural Network (Cannistra et al., 2021; John et al., 2022) models.

While we agree that some background on each machine model model is important, there is expansive literature on each of these machine learning models and mathematical definitions are easily accessible through the SciKit Learn documentation and elsewhere. Because our work is focused on the application rather than development or augmentation of the machine learning models, we will point the reader to resources where they can learn more about the models if they so choose L204:
"For more information on the mathematical basis and implementation of each machine learning model, refer to the SciKit Learn documentation (https://scikit-learn.org/stable/user_guide.html)."
As suggested by Reviewer 2 below, we will include the hyperparameters used for each model in the Supplementary Material and refer to that material as well.

Title – Consider including machine learning and optical imagery in the title, such as, 'Automated snow cover detection on mountain glaciers using optical imagery AND machine learning'.

We will modify the title according to your suggestion.

Figure 1 – Include caption for Figure (d).

Thank you for catching our mistake. After making edits to panels c and d, we will make sure that all panels are fully described in the caption.

Figure 2 – Include grid or scale bar/northing arrow and land boundary or better contrast in colors between land and ocean in the location image. Increase font size/bold labels.

We will implement all of your suggestions: include grid and northing/easting coordinates, increase contrast in the basemap, and increase the font size/bold labels.

Line 150 – Figure 3 → should be Figure 4 and order of figures needs to be reconsidered within the text.
Thank you for catching this. We will make sure all figures are labeled and referenced correctly in the next version.

Line 165 – Can you include the process of training and validation dataset development within figure 4? Or a separate figure to have a visual summary of how the datasets were developed.
Great idea, we will add an additional figure or panel to Figure 4 to demonstrate the model training, testing, and validation datasets construction.

Line 165 – This contradicts Line 270 where it mentions that Lemon Creek Glacier was used in the performance assessment, should that also be mentioned here?
You are right, thank you for catching this mistake. We will mention here that Lemon Creek was only included in the performance assessment. The section will also be modified to better separate the training, testing, and validation sets, with support from the new figure you suggested.

Line 167 – Consider listing the nine ML models here or state that they will be listed at the end of this section.
We will mention that the models are described below.

Line 169 – Include number of folds here.
We will add the number of folds here.

Line 172 – 188 – This paragraph could be better clarified to first talk about the validation dataset and then what's used for training. Some of the information is contained within the preceding paragraph, so it is difficult to know if it is a repeat or new information.
Great point, discussion of the training, testing, and validation datasets is not organized very clearly. We will restructure this section as follows to better align with the new figure and Figure 4:
2.1. Training, testing, and validation datasets construction
2.2. Image pre-processing
2.3. Classification model development and application
2.4. Snowline detection

Line 189 – Did this result in an even number of points for each class? If not, how do you reconcile the bias within classes that are under sampled?
Great question. The sample points used for model testing and training were slightly biased towards snow-covered pixels compared to other classes, yet we found this had little impact on the classification accuracies/results. We will make this clear in the text by adding the following to L189:
"This sampling method led to a training dataset with more snow-covered points compared to other classes due to the larger relative area of snow in each image particularly early in the melt season. Therefore, we tested several configurations of the

training dataset (e.g., stratified proportional sampling) and found little to no impact on the classification accuracies and results."

In the validation dataset, the snow-covered and snow-free sample distributions were more similar (~1400 and ~1300, respectively), so we will make this clear in the validation dataset description (new section 2.1).

Line 204 – Was this at all sites for each image product?
Yes, each model was applied to the full training and testing set, which included four sites: Gulkana, Wolverine, South Cascade, and Sperry Glaciers. We will clarify this with the following:
"For each image product, we tested nine machine learning models on the respective training dataset…"

Line 215 – Mention that this is expanded on in supplementary material.
At the end of this sentence, we will add "...detailed in the Supplementary Material (S2)."

Line 247 – Clarify that you are discussing situations for missing data.
In this section, we are discussing how we filled the SCA so that the snowline is not detected in the middle of the SCA where there are small rocks or bare ice for example, rather than at the lowermost and longest snow-ice boundary. To clarify, we will modify this sentence:
"To prevent the snowline from being detected within the SCA, such as at areas of exposed bedrock or crevasses, or at small patches of snow outside the SCA, classified images were adjusted using the distribution of snow-covered pixels and the boundaries between snow and no-snow ("edges")."

Line 343 – Refer to either Table 2 or Figure 7 to where to look for the results of changes in snow line detection.
We will refer to Table 2 here, thank you for the suggestion.

Line 360 – SCA has been defined earlier.
We will replace "snow-covered area (SCA)" with "SCA" here.

Line 366 – Refer to figure 5 (a-f) here, as is done in the next paragraph.
We will modify this sentence to the following:
"In general, the transient AAR time series suggest that the largest, most northerly sites, Gulkana and Wolverine Glaciers (Fig. 5b,e), have higher annual AARs compared to other sites (Fig. 5h,k,n)."

Line 401 – Consider renaming this section to 'Challenges in Classification' or something similar, since limitations are not necessarily being discussed here.
Good idea, we will rename this section "Snow detection challenges" to better summarize the topics.

Line 505 – Readers should be referred to Table S2 for specifics on when glacier boundaries were updated.
We will reference Table S2 in this sentence: "If glacier boundaries are not updated over time as in this study (Table S2), the AAR will be underestimated."

Figure 6 – Include in caption what the white arrows are highlighting.
Good idea, in the Figure 6 caption, we will add "...as indicated by the white arrows" where appropriate.

Line 510 – This section does not mention future work. Consider changing title or including future work.
Thank you for pointing this out. We will change this section title to "Broader implications."

Line 570 – Mention in the methods section that images were accessed with GEE.
We will add the following to L138: "...we developed the snow detection workflow using Landsat 8/9, PlanetScope, and Sentinel-2 imagery, accessed through the Google Earth Engine data repository."

Supplemental material:
Line 601 – Check figure numbering, Figure S4 → S3
Thank you, we will make sure all figures are labeled and referenced correctly in the Supplementary Material.

**Review 2**
Major comments:
1. The introduction does a good job of presenting the benefits of and need for improved snow detection methods. However, it should be expanded to include greater discussion and acknowledgement of the existing body of research that has focused on snow identification on glacier surfaces.
We agree, the need for more discussion of previous snow classification on glaciers was also pointed out by Reviewer 1, so we will make sure to include this. Please see the response to the first major comment from Reviewer 1.

2. Individual portions of the methods section are well written, but I suggest reorganizing it to make the entire story easier to follow. Specifically: 1) the study area should be moved to its own section, rather than included in the methods, and 2) Sections 2.2-2.4 could be reorganized to follow the structure of Figure 4 to make it easier for the reader to follow. For example, large parts of imagery selection and pre-processing are currently included in Section 2.4 but may fit better in section 2.2, and the seasonal snowline identification could be broken out into its own section.
Thank you for the suggestions. We will move the study area to its own section. Reviewer 1 also noted some confusion with the organization in this section, so we will make several changes. See the response to the comment from Reviewer 1 above starting with "Line 172 – 188".

3. The development of reproduceable and extendable code/methods is an important aspect of this project, however I find that the areas of the manuscript where the authors discuss details of the code to be confusing and distracting (e.g. lines 226-227, 234-235). I would encourage the authors to consider whether these details are better suited to be included in the supplemental information or as details on the github page.

We appreciate the feedback on discussing specific code parameters and settings. We will remove these details for readability.

4. The authors have put considerable effort into developing the methods and creating a thorough dataset. However, I think that more space should be used to present details of the derived products. For example, Figure S3 contains many useful insights that would be better suited for the main manuscript. Specifically, it highlights the differences in temporal resolution between the different imagery products, and well as how consistent (or inconsistent in the case of PlanetScope imagery) the derived products are. Highlighting these results in a main-text figure would improve the presentation of the findings (or perhaps a subset of this figure, such as only a single glacier, or only a subset of years, such that the details of the plot are more easily seen). Other questions which are raised in this figure and throughout the manuscript which could be elaborated on include: are you able to identify significant interannual-variability in the glacier snowline elevation and AAR from these products? How would the results compare when using only a single imagery source, rather than a blend of all imagery as you have done here?

Thank you for your suggestions regarding Figure S3 and potential expansions of the discussion. We will move Figure S3 to the main text and provide more discussion in Section 4.2 on the derived products. The noisy PlanetScope time series are more apparent in Figure S3, so it would be valuable to reference the figure where this is discussed in L464. We agree that the ability to capture interannual variability in AARs is important, so we will point this out in the text, as demonstrated for Lemon Creek in particular (Figure S3c). Regarding the use of different image sources, moving Figure S3 will also allow us to discuss how Landsat 8/9 has varying importance on the time series, depending on the site and roughly correlated to the site size. For example, Landsat 8/9 provides more frequent observations at the Alaskan glaciers, particularly before 2016, while there are fewer Sentinel-2 images overall. However, for the smaller, lower latitude glaciers (Sperry and South Cascade Glaciers), Sentinel-2 images are more abundant and Landsat images are sparse for the full study period. We will add these observations to L472–480 where appropriate.

Minor comments:
Line 68: has -> have
We will fix this.

Line 69: I would suggest rephrasing "images with spatial resolutions of 1 km or more" to remove the specific number, as most commonly-used satellite imagery is finer spatial resolution.
Good idea, we will rephrase this as you have suggested.

Line 92: I found that these two points (particularly point 1) were difficult to read. You might consider simplifying or restructuring the sentence here.
Thank you for the feedback, we will rephrase these sentences as follows:
"Our goals in this work are two-fold: (1) Develop an automated snow detection workflow calibrated to glacier surfaces by evaluating several machine learning algorithms, and (2) compare the results from individual image products and snow cover metrics to assess spatiotemporal trends in glacier snow cover."

Line 109: It should be clarified that the manually generated snow cover observation were made from satellite imagery, rather than from in situ observations.
We will specify "from satellite imagery" here.

Line 132: It was a bit confusing to see Emmons Glacier included in this figure immediately after the study area section, where it was not mentioned. Perhaps the details on how it is used should be included earlier in the manuscript to avoid this confusion.
Good idea, we will include a description of Emmons Glacier in the Study Sites section with a brief justification for its inclusion in the study. We also hope that describing the training/testing and validations datasets in their own section (new 2.1) will help with this confusion.

Line 147: The reference to Figure 3 should be to Figure S1, I believe.
Thank you for catching this, we meant to reference Figure 4 here which includes the "Adjust radiometry" step for PlanetScope. We will correct this and check all other figure references.

Line 204: The inclusion of nine separate ML models is impressive and thorough. Additional information should be included for each (likely in the supplement, I would think) on the specific hyperparameters used for each.
Great point, we will include a section on the hyperparameters used for all of the machine learning models in the Supplementary Material.

Line 252: How are the masked areas treated in the process of making these histogram? Are the masked pixels included in the glacier elevation bin histogram?
All pixels are included in both the snow-covered and elevation histograms for the purpose of snowline detection. However, masked pixels are not used for actual snowline detection. In other words, masked pixels can be used to remove potential snowlines but not to identify them. We will make this more clear in the text by modifying L254: "...all elevation bins with at least 75% snow coverage were set to 100%, and the image pixels, including cloud-masked pixels in the glacier area, were adjusted accordingly.

Line 257: What is included in the no-data mask here? Is it only cloudy pixels? Cloudy pixels and off-glacier areas?
Thank you for bringing up an important point of clarification. In L257, we will add the following: "The no data mask associated with the binary snow image includes all pixels

outside the glacier area and cloud-masked pixels not filled in the previous histogram-based filling step."

Line 254-255: I worry that this may cause a consistent negative bias in the snowline altitudes which are derived. Was a similar approach used to remove sparse snow patches at low elevations to ensure that these snow-ice boundaries were not included?
Good question. The elevation range and length filters (L259–260) were applied mainly to remove low-elevation snowlines. We found these filters to be effective for isolating the largest snow-ice boundary on the glacier, which was our general approach, so we will make this clearer in the text. Other approaches would be an interesting comparison in the future, and could potentially pose questions about which is the most appropriate "snowline" in cases where there are several snow-ice boundaries across a wide elevation range.

Line 340: typo for "instils"
"Instill" is used predominantly over "instil" in the U.S., so we will keep this spelling for consistency with spelling conventions in the rest of the text.

Line 343: How are the differences in timing of the manual vs automated snowlines treated in this comparison? What is the range of differences?
The manual and automated snowlines were compared for the same images, so there is no time difference between observations. We will clarify this by adding "for a given image" on L343.

Line 343-352: I don't think the +/- symbol should be used for the IQR numbers here. Including the actual min/max of the IQR would be a more useful metric. eg "… differ from manually delineated snowlines by a median of 116 m (IQR 20–259 m) in ground distance …"
We agree with your suggestion. All median +/- IQR mentions will be changed to median of XX m (IQR of XX m).

Line 344: including a figure (scatterplot) showing the relationship between automated vs manually-delineated snowline altitude would be a useful addition to highlight the accuracy of the automated methods.
Good idea, we will add a figure showing the automated vs. manual snowline altitudes scatterplot to Section 3.1.

Line 370: Is "the ranges in transient AARs are much larger (0.1–0.3)" referring to interannual ranges in AAR, intra-annual, or range amongst the glaciers?
Thank you for pointing out this confusing sentence. To simplify, we will replace it with: "In comparison, the average AARs are lower at Lemon Creek (~0.1–0.4), Sperry (~0.5), and South Cascade (0.2–0.4) Glaciers."

Line 425: I believe this is the first time that cloud shadows are discussed. The methods should be more explicit that cloud shadows are identified and removed from the imagery.

Thank you for pointing this out. We mentioned this in the Methods (L222), but will add details there about the specific cloud masking parameters and reference the package used for automated cloud masking (geedim).

Lines 440 & 455: I was initially a little confused by these statements on more heavily weighting Sentinel-2 SR observations. I would suggest rephrasing these to make it more clear that the suggestion is for when observations from multiple sources are being synthesized.
For clarity here, we will add "when combining Sentinel-2 observations with those from other image products" or similar to both of the lines that you noted.

Supplement Line 72: typo (repeated words)
This will be fixed.

Figure 1: A note should be made in the caption (and/or an asterisk added) to acknowledge that the NDSI bands indicated for PlanetScope imagery is not the typically-used SWIR band.
We will add the PlanetScope bands used for NDSI to the caption as suggested.

Figure 2: I find that having only the elevation makes the setting of these glaciers difficult to interpret. I would suggest including a background hillshade on each glacier to accentuate the local topography, and perhaps use a colormap with more breaks to better highlight changes in elevation (such as the matplotlib "terrain" or "gist_earth" colormaps). The authors may also consider removing the Easting/Northing grid labels from each panel and instead include an inset scale bar for each, to allow more space for each panel to be larger.
Thank you for the suggestions. To each of the maps, we will change the colormap to "terrain," add underlying hillshades, and replace the grid labels with an inset scale bar for easier interpretation.

Figure 5: I don't feel that it is necessary to include both SCA and AAR here, as the patterns of each are nearly identical. I personally find the AAR to be a more useful metric in this visualization, as it allows direct comparison between the glaciers.
We agree with your suggestion. We will remove the SCA panel here and instead include Figure S3 in the main text, for reasons mentioned above.

Figure S3: I find it difficult to tell the difference between the Sentinel-2 SR and TOA markers. Could a different color or shape be used to better highlight the difference between them?
Good idea, we will change the Sentinel-2 marker types and colors so that they are more easily distinguishable.

**References**
Callegari, M., Carturan, L., Marin, C., Notarnicola, C., Rastner, P., Seppi, R., & Zucca, F. (2016).

A Pol-SAR Analysis for Alpine Glacier Classification and Snowline Altitude Retrieval.

*IEEE Journal of Selected Topics in Applied Earth Observations and Remote Sensing*, *9*(7), 3106–3121. https://doi.org/10.1109/JSTARS.2016.2587819

Cannistra, A. F., Shean, D. E., & Cristea, N. C. (2021). High-resolution CubeSat imagery and machine learning for detailed snow-covered area. *Remote Sensing of Environment*, *258*, 112399. https://doi.org/10.1016/J.RSE.2021.112399

Dozier, J. (1989). Spectral signature of alpine snow cover from the landsat thematic mapper. *Remote Sensing of Environment*, *28*, 9–22. https://doi.org/10.1016/0034-4257(89)90101-6

Gascoin, S., Grizonnet, M., Bouchet, M., Salgues, G., & Hagolle, O. (2019). Theia Snow collection: High-resolution operational snow cover maps from Sentinel-2 and Landsat-8 data. *Earth System Science Data*, *11*(2), 493–514. https://doi.org/10.5194/essd-11-493-2019

Hall, D. K., & Riggs, G. A. (2007). Accuracy assessment of the MODIS snow products. *Hydrological Processes*, *21*(12), 1534–1547. https://doi.org/10.1002/hyp.6715

Huang, L., Li, Z., Tian, B., Chen, Q., & Zhou, J. (2013). Monitoring glacier zones and snow/firn line changes in the Qinghai–Tibetan Plateau using C-band SAR imagery. *Remote Sensing of Environment*, *137*, 17–30. https://doi.org/10.1016/j.rse.2013.05.016

John, A., Cannistra, A. F., Yang, K., Tan, A., Shean, D., Hille Ris Lambers, J., & Cristea, N. (2022). High-Resolution Snow-Covered Area Mapping in Forested Mountain Ecosystems Using PlanetScope Imagery. *Remote Sensing*, *14*(14), 3409. https://doi.org/10.3390/rs14143409

Li, Z., Huang, L., Chen, Q., & Tian, B. (2012). Glacier Snow Line Detection on a Polarimetric SAR Image. *IEEE Geoscience and Remote Sensing Letters*, *9*(4), 584–588. https://doi.org/10.1109/LGRS.2011.2175697

Otsu, N. (1979). A Threshold Selection Method from Gray-Level Histograms. *IEEE Transactions on Systems, Man, and Cybernetics*, *SMC-9*(1), 62–66.

Prieur, C., Rabatel, A., Thomas, J.-B., Farup, I., & Chanussot, J. (2022). Machine Learning

    Approaches to Automatically Detect Glacier Snow Lines on Multi-Spectral Satellite

    Images. *Remote Sensing*, *14*(16), 3868. https://doi.org/10.3390/rs14163868

Rastner, P., Prinz, R., Notarnicola, C., Nicholson, L., Sailer, R., Schwaizer, G., & Paul, F. (2019).

    On the Automated Mapping of Snow Cover on Glaciers and Calculation of Snow Line

    Altitudes from Multi-Temporal Landsat Data. *Remote Sensing*, *11*(12), 1410.

    https://doi.org/10.3390/rs11121410

Riggs, G. A., Hall, D. K., & Salomonson, V. V. (1994). A snow index for the Landsat Thematic

    Mapper and Moderate Resolution Imaging Spectroradiometer. In *Proceedings of

    IGARSS '94 - 1994 IEEE International Geoscience and Remote Sensing Symposium*

    (Vol. 4, pp. 1942–1944 vol.4). https://doi.org/10.1109/IGARSS.1994.399618

Salomonson, V. V., & Appel, I. (2004). Estimating fractional snow cover from MODIS using the

    normalized difference snow index. *Remote Sensing of Environment*, *89*(3), 351–360.

    https://doi.org/10.1016/j.rse.2003.10.016

Sankey, T., Donald, J., McVay, J., Ashley, M., O'Donnell, F., Lopez, S. M., & Springer, A. (2015).

    Multi-scale analysis of snow dynamics at the southern margin of the North American

    continental snow distribution. *Remote Sensing of Environment*, *169*, 307–319.

    https://doi.org/10.1016/j.rse.2015.08.028

Zeller, L., McGrath, D., Sass, L. C., Florentine, C. E., & Downs, J. (In review). Equilibrium line

    altitudes, accumulation areas, and the vulnerability of glaciers in Alaska. *Journal of

    Glaciology*.

---

## Author Response (AR1)

We thank the reviewers for their thoughtful comments and suggestions for the manuscript. Below, we outline our revisions based on the comments from each reviewer. Reviewer comments are shown in blue and our responses are shown in black.

**Review 1**

Snowline detection is an illustrious metric within glaciology for first order estimates of surface mass balance and for understanding melt extents/intensity. Published research in this area of remote sensing has been somewhat extensive but that is not reflected in the introduction or discussion of the paper. The authors should rewrite the introduction and discussion to reference relevant literature on machine learning and snowline detection.

We agree that more background on glacier snow classification and machine learning in particular is needed in the Introduction. To better couch this work in the literature related to classifying glacier snow cover with a focus on machine learning, we expanded the discussion of the Normalized Difference Snow Index (NDSI) and its shortcomings to other snow mapping thresholding and machine learning techniques in the modified Introduction (L61–L88). Thresholding techniques include the NDSI (Dozier, 1989; Gascoin et al., 2019; Hall & Riggs, 2007; Riggs et al., 1994; Salomonson & Appel, 2004; Sankey et al., 2015) and Otsu thresholding, an automated threshold selection approach for gray-level images (Prieur et al., 2022; Rastner et al., 2019) (Prieur et al., 2022). For machine learning techniques, we discuss Neural Networks applied to PlanetScope (Cannistra et al., 2021; John et al., 2022), Random Forest applied to Sentinel-2 imagery (Zeller et al., In review), and Support Vector Machine applied to C-band SAR imagery (Callegari et al., 2016; Huang et al., 2013; Li et al., 2012).

We also modified Figure 1 to better support these changes. Instead of the NDSI-based classification panels, we present box plots of band reflectance and NDSI values in Sentinel-2 images captured throughout several melt seasons for sample snow-, ice-, and firn-covered pixels. We believe this better illustrates one of the main challenges of automated snow and ice classification: the overlapping and temporally evolving reflectances of snow, ice, and firn.

The methods section gives a good overview of each step and is nicely summarized in Figure 4. I would like to see additional sections that expands on the machine learning models, giving a brief background on the nine different types and why they are selected. For instance, identifying which models are ensemble tree types, kernel type, neural type. Perhaps, highlighting which models have been previously applied in glacier applications.

To provide more context for each of the machine learning models tested, we edited L222:

"For each image product, we tested nine supervised ML models, including linear (Logistic Regression, Nearest Neighbors), quadratic (Quadratic Discriminant Analysis), non-parametric (Decision Tree), kernel-based (Support Vector Machine), ensemble (AdaBoost, Random Forest), Naïve Bayes, and Neural Network models."

While we agree that some background on each machine model is important, there is expansive literature on each of these machine learning models and mathematical definitions are easily accessible through the Scikit-Learn documentation and elsewhere. Because our work is focused on the application rather than development or augmentation of the machine learning models, we now point the reader to resources where they can learn more about the models if they so choose (L230):
"For more information on the mathematical basis and implementation of each machine learning model, refer to the Scikit-Learn documentation (https://scikit-learn.org/stable/user_guide.html)."

As suggested by Reviewer 2 below, we also included the hyperparameters used for each model in the supplement (S2; Table S1).

Title – Consider including machine learning and optical imagery in the title, such as, 'Automated snow cover detection on mountain glaciers using optical imagery AND machine learning'.
We modified the title according to your suggestion.

Figure 1 – Include caption for Figure (d).
Thank you for catching our mistake. All panels are now described in the caption.

Figure 2 – Include grid or scale bar/northing arrow and land boundary or better contrast in colors between land and ocean in the location image. Increase font size/bold labels.
We implemented all of your suggestions: included grid and northing/easting coordinates, increased contrast in the basemap, and increased the font size/bold labels.

Line 150 – Figure 3 → should be Figure 4 and order of figures needs to be reconsidered within the text.
Thank you for catching this. We checked that all figure labels and references are now correct.

Line 165 – Can you include the process of training and validation dataset development within figure 4? Or a separate figure to have a visual summary of how the datasets were developed.
Great idea, we added an additional figure to demonstrate the model training, testing, and validation datasets construction (new Fig. 4).

Line 165 – This contradicts Line 270 where it mentions that Lemon Creek Glacier was used in the performance assessment, should that also be mentioned here?
You are right, thank you for catching this mistake. We corrected this in the text.

Line 167 – Consider listing the nine ML models here or state that they will be listed at the end of this section.
We added "listed in 3.3" to this sentence.

Line 169 – Include number of folds here.
We added the number of folds here.

Line 172 – 188 – This paragraph could be better clarified to first talk about the validation dataset and then what's used for training. Some of the information is contained within the preceding paragraph, so it is difficult to know if it is a repeat or new information.
Great point, discussion of the training, testing, and validation datasets were not organized very clearly. To address this confusion, we restructured the Methods section as follows:
2.1. Training, testing, and validation datasets construction
2.2. Image pre-processing
2.3. Classification model development and application
2.4. Snowline detection
2.5. Performance assessment

Line 189 – Did this result in an even number of points for each class? If not, how do you reconcile the bias within classes that are under sampled?
The sample points used for model testing and training were slightly biased towards snow-covered pixels compared to other classes, yet we found this had little impact on the classification accuracies/results. We added the following to address your question (L159):
"This sampling method led to the most snow-covered points due to the larger relative area of snow early in the melt season. We tested several configurations of the training dataset (e.g., stratified proportional sampling) and found little to no impact on the classification accuracies and results."

Line 204 – Was this at all sites for each image product?
Yes, each model was applied to the full training/testing dataset. To clarify, we added the following to this sentence (L222):
"For each image product, we trained and tested nine supervised ML models on the respective training dataset…"

Line 215 – Mention that this is expanded on in supplementary material.
At the end of this sentence, we added "...detailed in the supplement (S4; Fig. S2)."

Line 247 – Clarify that you are discussing situations for missing data.
In this section, we are discussing how we filled the SCA so that the snowline is not detected in the middle of the SCA where there are small rocks or bare ice for example, rather than at the lowermost and longest snow-ice boundary. To clarify, we rephrased this sentence (L256):
"To prevent the snowline from being detected within the SCA, such as at areas of exposed bedrock or crevasses, or at small patches of snow, classified images were adjusted…"

Line 343 – Refer to either Table 2 or Figure 7 to where to look for the results of changes in snow line detection.

We now refer to Table 2 here, thank you for the suggestion.

Line 360 – SCA has been defined earlier.
We replaced "snow-covered area (SCA)" with "SCA" here and at all later mentions.

Line 366 – Refer to figure 5 (a-f) here, as is done in the next paragraph.
We modified this sentence to the following (L399):
"In general, the transient AAR time series suggest that the largest, most northerly sites, Gulkana and Wolverine Glaciers (Fig. 8b, e), have higher annual AARs compared to other sites (Fig. 8h, k, n)."

Line 401 – Consider renaming this section to 'Challenges in Classification' or something similar, since limitations are not necessarily being discussed here.
Good idea, we renamed this section "Snow detection challenges" to better summarize the topics.

Line 505 – Readers should be referred to Table S2 for specifics on when glacier boundaries were updated.
We now reference the appropriate table in this sentence (L545): "If glacier boundaries are not updated over time as in this study (Table S2), the AAR will be underestimated."

Figure 6 – Include in caption what the white arrows are highlighting.
Good idea, in the figure caption (now Figure 9), we added "...as indicated by the white arrows" where appropriate.

Line 510 – This section does not mention future work. Consider changing title or including future work.
Thank you for pointing this out. We changed this section title to "Broader implications."

Line 570 – Mention in the methods section that images were accessed with GEE.
We added the following to L195: "We accessed Landsat and Sentinel-2 images through the Google Earth Engine data repository and Sentinel-2-harmonized PlanetScope 4-band Surface Reflectance Scenes through the Planet Labs, Inc. Python API."

Supplemental material:
Line 601 – Check figure numbering, Figure S4 → S3

Thank you, we made sure all figures are labeled and referenced correctly in the Supplementary Material.

**Review 2**
Major comments:
1. The introduction does a good job of presenting the benefits of and need for improved snow detection methods. However, it should be expanded to include greater discussion and acknowledgement of the existing body of research that has focused on snow identification on glacier surfaces.

We agree, the need for more discussion of previous snow classification on glaciers was also pointed out by Reviewer 1, so we modified parts of the introduction to address this. Please see the response to the first major comment from Reviewer 1.

2. Individual portions of the methods section are well written, but I suggest reorganizing it to make the entire story easier to follow. Specifically: 1) the study area should be moved to its own section, rather than included in the methods, and 2) Sections 2.2-2.4 could be reorganized to follow the structure of Figure 4 to make it easier for the reader to follow. For example, large parts of imagery selection and pre-processing are currently included in Section 2.4 but may fit better in section 2.2, and the seasonal snowline identification could be broken out into its own section.

Thank you for the suggestions. We moved the study area to its own section. Reviewer 1 also noted some confusion with the organization in this section, so we made several changes. See the response to the comment from Reviewer 1 above starting with "Line 172 – 188".

3. The development of reproduceable and extendable code/methods is an important aspect of this project, however I find that the areas of the manuscript where the authors discuss details of the code to be confusing and distracting (e.g. lines 226-227, 234-235). I would encourage the authors to consider whether these details are better suited to be included in the supplemental information or as details on the github page.

We appreciate the feedback on discussing specific code parameters and settings. We removed these details for readability.

4. The authors have put considerable effort into developing the methods and creating a thorough dataset. However, I think that more space should be used to present details of the derived products. For example, Figure S3 contains many useful insights that would be better suited for the main manuscript. Specifically, it highlights the differences in temporal resolution between the different imagery products, and well as how consistent (or inconsistent in the case of PlanetScope imagery) the derived products are. Highlighting these results in a main-text figure would improve the presentation of the findings (or perhaps a subset of this figure, such as only a single glacier, or only a subset of years, such that the details of the plot are more easily seen). Other questions which are raised in this figure and throughout the manuscript which could be elaborated on include: are you able to identify significant interannual-variability in the glacier snowline elevation and AAR from these products? How would the results compare when using only a single imagery source, rather than a blend of all imagery as you have done here?

Thank you for your suggestions regarding Figure S3 and potential expansions of the discussion. We moved Figure S3 to the main text (new Fig. 7) to more transparently discuss the noisy PlanetScope time series, the potential for capturing interannual variability in minimum SCA, and the varying impact of Landsat imagery on the density of the time series.

Minor comments:

Line 68: has -> have
We fixed this.

Line 69: I would suggest rephrasing "images with spatial resolutions of 1 km or more" to remove the specific number, as most commonly-used satellite imagery is finer spatial resolution.
We rephrased this as you suggested.

Line 92: I found that these two points (particularly point 1) were difficult to read. You might consider simplifying or restructuring the sentence here.
Thank you for the feedback, we rephrased these sentences as follows:
"Our goals in this work are two-fold: (1) Develop an automated snow detection workflow calibrated to glacier surfaces by evaluating several machine learning algorithms, and (2) compare the results from individual image products and snow cover metrics to assess the potential for capturing spatiotemporal trends in glacier snow cover."

Line 109: It should be clarified that the manually generated snow cover observation were made from satellite imagery, rather than from in situ observations.
We specified "from satellite imagery" here.

Line 132: It was a bit confusing to see Emmons Glacier included in this figure immediately after the study area section, where it was not mentioned. Perhaps the details on how it is used should be included earlier in the manuscript to avoid this confusion.
Good idea, we included a description of Emmons Glacier in the Study Sites section with a brief justification for its inclusion in the study. We hope that describing the training/testing and validations datasets in their own section (new 3.1) also helps with this confusion.

Line 147: The reference to Figure 3 should be to Figure S1, I believe.
Thank you for catching this, we meant to reference Figure 4 here which includes the "Adjust radiometry" step for PlanetScope. We corrected this and checked all other figure references.

Line 204: The inclusion of nine separate ML models is impressive and thorough. Additional information should be included for each (likely in the supplement, I would think) on the specific hyperparameters used for each.
Great point, we included a section on the hyperparameters used for all of the machine learning models in the supplement (S2, Table S1).

Line 252: How are the masked areas treated in the process of making these histogram? Are the masked pixels included in the glacier elevation bin histogram?
All pixels are included in both the snow-covered and elevation histograms for the purpose of snowline detection. However, masked pixels are not used for actual snowline detection. In other words, masked pixels can be used to remove potential snowlines but not to identify them. We clarified this in the text by modifying L265: "...all

elevation bins with at least 75% snow coverage were set to 100%, and the image pixels, including cloud-masked pixels in the glacier area, were adjusted accordingly.

Line 257: What is included in the no-data mask here? Is it only cloudy pixels? Cloudy pixels and off-glacier areas?
Thank you for bringing up an important point of clarification. In L268, we added the following: "The no-data mask associated with the binary snow image includes all pixels outside the glacier area and cloud-masked pixels not filled in the previous histogram-based filling step."

Line 254-255: I worry that this may cause a consistent negative bias in the snowline altitudes which are derived. Was a similar approach used to remove sparse snow patches at low elevations to ensure that these snow-ice boundaries were not included?
Good question. The elevation range and length filters were applied mainly to remove low-elevation snowlines. To clarify this, we added "To prevent the snowline from being detected within the SCA, such as at areas of exposed bedrock or crevasses, or at small patches of snow outside the SCA…" at the start of this sentence. We found these filters to be effective for isolating the largest snow-ice boundary on the glacier, which was our general approach (now stated in L256).

Line 340: typo for "instils"
"Instill" is used predominantly over "instil" in the U.S., so we will keep this spelling for consistency with spelling conventions in the rest of the text.

Line 343: How are the differences in timing of the manual vs automated snowlines treated in this comparison? What is the range of differences?
The manual and automated snowlines were compared for the same PlanetScope images and for Landsat and Sentinel-2 images within one week of the respective PlanetScope image. We now include this point on L322:
"Automatic snowlines for Landsat and Sentinel-2 imagery were only compared when the image capture date was within one week of the respective PlanetScope image."

Line 343-352: I don't think the +/- symbol should be used for the IQR numbers here. Including the actual min/max of the IQR would be a more useful metric. eg "… differ from manually delineated snowlines by a median of 116 m (IQR 20–259 m) in ground distance …"
We agree with your suggestion. All median +/- IQR mentions were changed to median of XX (IQR of XX to XX).

Line 344: including a figure (scatterplot) showing the relationship between automated vs manually-delineated snowline altitude would be a useful addition to highlight the accuracy of the automated methods.
Good idea, we added a figure showing the automated vs. manual snowline altitudes scatterplot to Section 4.1 (new Figure 6).

Thank you for pointing out this confusing sentence. To simplify, we replaced it with: "In comparison, the average AARs are lower at Lemon Creek (~0.1–0.4), Sperry (~0.5), and South Cascade (0.2–0.4) glaciers."

We previously mentioned this in the Methods, but added details about the specific cloud masking parameters and reference the package used for automated cloud masking (geedim) in L200.

For clarity here, we added "when combining Sentinel-2 observations with those from other image products" or similar to both of the lines that you noted.

This was fixed.

We added the PlanetScope bands used for NDSI to the caption as suggested.

Thank you for the suggestions. To each of the maps, we changed the colormap, added underlying hillshades, and replaced the grid labels with an inset scale bar for easier interpretation.

We agree with your suggestion. We removed the SCA panel here and instead included Figure S3 in the main text, for reasons mentioned above.

**Figure S3: I find it difficult to tell the difference between the Sentinel-2 SR and TOA markers. Could a different color or shape be used to better highlight the difference between them?**

We changed the Sentinel-2 marker types so that they are more easily distinguishable.

**References**

Callegari, M., Carturan, L., Marin, C., Notarnicola, C., Rastner, P., Seppi, R., & Zucca, F.
(2016). A Pol-SAR Analysis for Alpine Glacier Classification and Snowline Altitude
Retrieval. *IEEE Journal of Selected Topics in Applied Earth Observations and Remote
Sensing*, *9*(7), 3106–3121. https://doi.org/10.1109/JSTARS.2016.2587819

Cannistra, A. F., Shean, D. E., & Cristea, N. C. (2021). High-resolution CubeSat imagery and
machine learning for detailed snow-covered area. *Remote Sensing of Environment*, *258*,
112399. https://doi.org/10.1016/J.RSE.2021.112399

Dozier, J. (1989). Spectral signature of alpine snow cover from the landsat thematic mapper.
*Remote Sensing of Environment*, *28*, 9–22. https://doi.org/10.1016/0034-
4257(89)90101-6

Gascoin, S., Grizonnet, M., Bouchet, M., Salgues, G., & Hagolle, O. (2019). Theia Snow
collection: High-resolution operational snow cover maps from Sentinel-2 and Landsat-8
data. *Earth System Science Data*, *11*(2), 493–514. https://doi.org/10.5194/essd-11-493-
2019

Hall, D. K., & Riggs, G. A. (2007). Accuracy assessment of the MODIS snow products.
*Hydrological Processes*, *21*(12), 1534–1547. https://doi.org/10.1002/hyp.6715

Huang, L., Li, Z., Tian, B., Chen, Q., & Zhou, J. (2013). Monitoring glacier zones and snow/firn
line changes in the Qinghai–Tibetan Plateau using C-band SAR imagery. *Remote
Sensing of Environment*, *137*, 17–30. https://doi.org/10.1016/j.rse.2013.05.016

John, A., Cannistra, A. F., Yang, K., Tan, A., Shean, D., Hille Ris Lambers, J., & Cristea, N.
(2022). High-Resolution Snow-Covered Area Mapping in Forested Mountain
Ecosystems Using PlanetScope Imagery. *Remote Sensing*, *14*(14), 3409.

https://doi.org/10.3390/rs14143409

Li, Z., Huang, L., Chen, Q., & Tian, B. (2012). Glacier Snow Line Detection on a Polarimetric

SAR Image. *IEEE Geoscience and Remote Sensing Letters*, *9*(4), 584–588.

https://doi.org/10.1109/LGRS.2011.2175697

Otsu, N. (1979). A Threshold Selection Method from Gray-Level Histograms. *IEEE Transactions

on Systems, Man, and Cybernetics*, *SMC-9*(1), 62–66.

Prieur, C., Rabatel, A., Thomas, J.-B., Farup, I., & Chanussot, J. (2022). Machine Learning

Approaches to Automatically Detect Glacier Snow Lines on Multi-Spectral Satellite

Images. *Remote Sensing*, *14*(16), 3868. https://doi.org/10.3390/rs14163868

Rastner, P., Prinz, R., Notarnicola, C., Nicholson, L., Sailer, R., Schwaizer, G., & Paul, F.

(2019). On the Automated Mapping of Snow Cover on Glaciers and Calculation of Snow

Line Altitudes from Multi-Temporal Landsat Data. *Remote Sensing*, *11*(12), 1410.

https://doi.org/10.3390/rs11121410

Riggs, G. A., Hall, D. K., & Salomonson, V. V. (1994). A snow index for the Landsat Thematic

Mapper and Moderate Resolution Imaging Spectroradiometer. In *Proceedings of

IGARSS '94 - 1994 IEEE International Geoscience and Remote Sensing Symposium*

(Vol. 4, pp. 1942–1944 vol.4). https://doi.org/10.1109/IGARSS.1994.399618

Salomonson, V. V., & Appel, I. (2004). Estimating fractional snow cover from MODIS using the

normalized difference snow index. *Remote Sensing of Environment*, *89*(3), 351–360.

https://doi.org/10.1016/j.rse.2003.10.016

Sankey, T., Donald, J., McVay, J., Ashley, M., O'Donnell, F., Lopez, S. M., & Springer, A.

(2015). Multi-scale analysis of snow dynamics at the southern margin of the North

American continental snow distribution. *Remote Sensing of Environment*, *169*, 307–319.

https://doi.org/10.1016/j.rse.2015.08.028

Zeller, L., McGrath, D., Sass, L. C., Florentine, C. E., & Downs, J. (In review). Equilibrium line

altitudes, accumulation areas, and the vulnerability of glaciers in Alaska. *Journal of

*Glaciology.*

---

## Author Response (AR2)

13 February 2025

We thank the reviewer and the editor for their consideration and time taken to review our manuscript. Because the manuscript was accepted as is for publication, we made no changes to the text. The order of the supplementary material was slightly changed to align with when they are referenced in the main text, and references were adjusted accordingly.

In response to the suggestion from the previous file verification process, we modified the color scheme and symbology in Figure 11 to be more colorblind-friendly using the Coblis tool. The content and layout of the figure itself is unchanged.

Please do not hesitate to contact us if you have any further questions or concerns regarding our paper and the files included in the final submission.

Sincerely,
Rainey Aberle